# Drivers of change in Peak Season Surface Ozone Concentrations and Impacts on Human Health over the Historical Period (1850-2014)

Steven T. Turnock[1,2], Dimitris Akritidis[3,4], Larry Horowitz[5], Mariano Mertens[6], Andrea Pozzer[3,7], Carly L. Reddington[8], Hantao Wang[9], Putian Zhou[10], and Fiona O'Connor[1,11]

[1]Met Office Hadley Centre, Exeter, UK
[2]University of Leeds Met Office Strategic (LUMOS) Research Group, University of Leeds, UK
[3]Max Planck Institute for Chemistry, Atmospheric Chemistry Department, Mainz, Germany
[4]Department of Meteorology and Climatology, School of Geology, Aristotle University of Thessaloniki, Thessaloniki, Greece
[5]NOAA Geophysical Fluid Dynamics Laboratory, Princeton, NJ, USA
[6]Deutsches Zentrum für Luft- und Raumfahrt (DLR), Institut für Physik der Atmosphäre, Oberpfaffenhofen, Germany
[7]Climate and Atmosphere Research Center, The Cyprus Institute, Nicosia, 1645, Cyprus
[8]Institute for Climate and Atmospheric Science (ICAS), School of Earth and Environment, University of Leeds, UK
[9]Department of Environmental Sciences and Engineering, University of North Carolina at Chapel Hill, Chapel Hill, North Carolina 27599, United States
[10]Institute for Atmospheric and Earth System Research (INAR), University of Helsinki, Finland
[11]Department of Mathematics and Statistics, Global Systems Institute, University of Exeter, Exeter, UK

**Correspondence:** Steven Turnock (steven.turnock@metoffice.gov.uk)

**Abstract.** Elevated concentrations of ozone at the surface can lead to poor air quality and increased risks to human health. There have been large increases in surface ozone over the historical period associated with socio-economic development. Here the change in peak season ozone (OSDMA8) is estimated for the first time using hourly surface ozone output from 3 CMIP6 models over the 1850 to 2014 period. Additional results are obtained from one model to quantify the impact from different drivers of ozone formation, including anthropogenic emissions of ozone and aerosol precursors, stratospheric ozone and climate change. The peak season ozone concentrations are used to calculate the risk to human health, in terms of the attributable fraction metric (the percentage of deaths from COPD associated with long-term exposure to elevated ozone concentrations). OSDMA8 concentrations are simulated to increase by more than 50% across northern mid-latitude regions over the historical period, mainly driven by increases in anthropogenic emissions of $NO_x$ and global $CH_4$ concentrations. Small contributions are made from changes in other anthropogenic precursor emissions (CO and non-$CH_4$ VOCs), aerosols, stratospheric ozone and climate change. The proportion of the global population exposed to OSDMA8 concentrations above the theoretical minimum risk exposure level (32.4 ppb), increased from <20% in 1855 to >90% in 2010. This has also increased the risk to human health mortality due to COPD from long-term ozone exposure by up to 20% across Northern Hemisphere regions in the present day. Like for OSDMA8 concentrations, the drivers of the increase in the ozone health risks are attributed mainly to changes in $NO_x$ and global $CH_4$. Fixing anthropogenic $NO_x$ emissions at 1850 values can eliminate the risk to human health from long-term ozone exposure in the near present-day period. Understanding the historical drivers of ozone concentrations and their risk to human health can help to inform the dvelopment of future pathways that reduce this risk.

## 1 Introduction

Ozone is a greenhouse gas in the troposphere and classified as a secondary air pollutant at the Earth's surface, influencing both the Earth's radiative balance and regional air quality (Szopa et al., 2021). It is formed by photochemical reactions involving nitrogen oxides ($NO_x$), carbon monoxide (CO) and volatile organics compounds (VOCs, including $CH_4$) (Lelieveld and Dentener, 2000). Perturbations to the emissions associated with these precursors from both anthropogenic and natural sources can therefore alter net ozone production rates. In addition, the formation of ozone can also be impacted by geographical location and meteorological conditions such as temperature and photolysis rates (Monks et al., 2015). Furthermore, ozone concentrations at the surface are also affected by chemical destruction, surface deposition processes (Mills et al., 2018), hemispheric transport (Liang et al., 2018) and the downward transport of ozone from the stratosphere (Stohl et al., 2003; Chen et al., 2024), stratosphere-troposphere exchange (STE). In air pollution episodes, exposure to elevated surface concentrations of ozone can lead to impacts on human health associated with respiratory diseases such as chronic obstructive pulmonary disease (COPD) (Pozzer et al., 2023). The Global Burden of Disease (GBD) study in 2019 estimated that long term exposure to concentrations of ozone resulted in 365,000 annual mortalities from COPD (Murray et al., 2020), with a recent revision by the GBD 2021 assessment to 490,000 annual mortalities in 2021 from COPD (GBD 2021 Risk Factors Collaborators, 2024). Therefore, ozone pollution represents a large current risk to the health of a population requiring an understanding of the factors controlling changes in concentrations.

Surface ozone concentrations are understood to have rapidly increased throughout the $20^{th}$ Century due to industrial development and increased anthropogenic emissions of ozone precursors. Even with the uncertainties surrounding early measurements of surface ozone at the end of the $19^{th}$ and beginning of the $20^{th}$ Century, ozone concentrations are observed to have increased across the temperate and polar regions of the Northern Hemisphere by 30 to 70% (approximately 10 to 16 ppb) since the start of the $20^{th}$ Century (Tarasick et al., 2019). Previous multi-model inter comparison exercises have simulated the historical change in annual or seasonal mean tropospheric and surface ozone concentrations. In both the $5^{th}$ and $6^{th}$ phase of the Coupled Model Intercomparison Project (CMIP5 and CMIP6), the historical change in annual mean global surface ozone concentrations from 1850 to 2000 was simulated to be in agreement at 10 to 11 ppb (Young et al., 2013; Turnock et al., 2020). Simulated historical changes in annual mean surface ozone were larger at 15 to 20 ppb across northern hemisphere mid latitude regions in both CMIP5 and CMIP6, although some regions like south and east Asia showed large inter-model diversity. However, in both phases of CMIP the simulated historical changes in surface ozone concentrations were in part dominated by the large uncertainty in the simulated pre-industrial values of 1850 (Young et al., 2013; Turnock et al., 2020). Evaluating historical

changes in surface ozone concentrations is limited by the availability and reliability of observations with a sufficiently long record. Sufficient observation data is available over the last five to six decades across regions within the northern mid-latitudes (North America, Europe and Japan) and shows that both CMIP5 and CMIP6 models underestimated the recent rapid increase in surface ozone concentrations that occurred over these regions (Young et al., 2018; Parrish et al., 2014, 2021). However, the CMIP6 model simulated long-term change in ozone concentrations shows good agreement when compared to the observed trend at only remote observation locations (Griffiths et al., 2021). Evaluation of the present-day simulation of ozone concentrations in CMIP5 (2000) and CMIP6 (2005 to 2014 mean) shows that both generations of chemistry climate models are able to represent the spatial and seasonal distributions of ozone but overestimate the absolute concentrations at the surface compared to ground-based observations collected as part of the Tropospheric Ozone Assessment Report (TOAR) (Young et al., 2018; Turnock et al., 2020).

Dedicated model simulations allow for the studies focusing on the drivers of historical ozone changes. Generally, the most important drivers of historical ozone changes have been increases of anthropgenic emissions, especially emissions of $NO_x$ (e.g. Lelieveld and Dentener, 2000; Fiore et al., 2009; Young et al., 2013; Turnock et al., 2018), with different emission sectors having different effects on ozone due to different trends and ozone production efficiency (Dahlmann et al., 2011; Mertens et al., 2024). Besides increases in $NO_x$ emissions, the increase of methane is also an important driver (Wild et al., 2012; Iglesias-Suarez et al., 2018; Morgenstern et al., 2018). Moreover, changes in aerosols (Xing et al., 2017), emissions of $N_2O$ and ozone depleting substances (ODS) drive changes in tropospheric ozone (Morgenstern et al., 2018; Iglesias-Suarez et al., 2018; Zeng et al., 2022). Changes in climate via temperature, water vapor and radiation, can also influence surface ozone concentrations with increases and decreases generally anticipated in polluted and unpolluted regions respectively (Schnell et al., 2016; Fortems-Cheiney et al., 2017; Fu and Tian, 2019; Zanis et al., 2022). Uncertainties of various processes, such as description of natural emissions which differ in various models, however, hinders a consistent quantification of ozone drivers among various models (Archibald et al., 2020a).

Until lately, global estimates of ozone-attributable health impacts were based on outputs from global atmospheric-chemistry models providing surface ozone concentrations in sufficient spatio-temporal distribution for ozone exposure assessments (Anenberg et al., 2010; Lim et al., 2012; Lelieveld et al., 2015; Fang et al., 2013; Silva et al., 2013; Forouzanfar et al., 2015; Silva et al., 2016; Cohen et al., 2017; Malley et al., 2017; Liang et al., 2018; Lelieveld et al., 2020). However, as part of the first phase of TOAR, Fleming et al. (2018) provided an assessment of recent changes in relevant ozone health metrics calculated from observations, showing recent reductions since 2000 across northern mid latitudes regions (North America, Europe and East Asia). More recently, the synergistic use of in-situ ozone measurements and model outputs through data fusion have made it possible to assess ozone-related health effects (Murray et al., 2020; Malashock et al., 2022) based on observational-based gridded ozone data sets (DeLang et al., 2021; Becker et al., 2023). Over the previous years different ozone exposure metrics were applied, according to the adopted coefficients from relevant cohort studies. The exposure response function (ERF) for ozone is actually a mathematical formula expressing the relative risk (RR) of a disease as a function of ozone abundance.

Pope et al. (2002) and Ostro et al. (2004) introduced a log-linear ERF based on epidemiological findings, while later a log-log ERF approach was proposed Cohen et al. (2005). In the GBD 2019 report (Murray et al., 2020) a log-linear ERF was used, introducing an ozone exposure metric based on the ozone season daily maximum 8-hour mixing ratio (OSDMA8). Malashock et al. (2022), using the observation-based gridded OSDMA8 data set of DeLang et al. (2021) and the GBD2019 approach for mortality calculations, reported 423,000 ozone-related deaths worldwide for the year 2019, accounting for all respiratory diseases. The review by Pozzer et al. (2023) showed that global estimates of mortality from long-term exposure to ozone have a large range from 142,000 per year from COPD to 1.3 million per year from all respiratory diseases. Although there are several studies on the projected changes of ozone-related health effects under different climate change and demographic scenarios (West et al., 2007; Silva et al., 2016; Turnock et al., 2023; Pozzer et al., 2023; Akritidis et al., 2024), studies on the health effects of ozone during the historical period are lacking. In particular, few studies have quantified the change in risk to human health and the drivers behind this change over this period.

Here we make use of hourly mean surface ozone concentrations from three CMIP6 models that conducted historical transient experiments; a time resolution that has not been made available in previous global multi-model inter comparison exercises. This has allowed us to calculate an ozone metric relevant to human health (OSDMA8) and explore the simulated changes in concentrations and impacts on human health over the entire CMIP6 historical period (1850 to 2014). In addition, we also use output from sensitivity experiments conducted by one of the CMIP6 models to explore the impact of changes in the different drivers of ozone formation (precursor emissions, stratospheric ozone, aerosols, climate) on the simulated concentrations and how this relates to changes in the risk to human health from long-term exposure to ozone over the same time period.

## 2 Methods

### 2.1 CMIP6 Model Data

We use output from the histSST experiments conducted by three different CMIP6 models as part of the Aerosols and Chemistry Model Intercomparison Project (AerChemMIP), a CMIP6 endorsed Model Intercomparison Project (Collins et al., 2017). The histSST experiments were designed to be an atmosphere-only representation of the coupled historical experiment conducted by individual models using historical forcing data over the period 1850 to 2014. The three models used here, UKESM1-0-LL (Sellar et al., 2019, 2020), GFDL-ESM4 (Horowitz et al., 2020; Dunne et al., 2020) and EC-Earth3-AerChem (van Noije et al., 2021), are all global Earth system models with horizontal grid areas coarser than 100 km. All three models include an interactive representation of chemistry and aerosols, with various different couplings and feedback within the Earth system e.g. land surface, radiation etc. All models used in this study prescribe long-lived greenhouse gases and methane as global annual concentrations. Table A1 shows a brief description of relevant chemistry and aerosols processes in the three different CMIP6 models. They are therefore suitable to use for simulating the long-term response to surface ozone from perturbations to different drivers of ozone formation, along with any feedbacks from changes in climate.

**Table 1.** Experiment configuration of the historical sensitivity experiments used in this study. The 3 CMIP6 models EC-Earth3-AerChem, GFDL-ESM4 and UKESM1-0-LL all provide data for histSST whereas, only UKESM1-0-LL provides data for the other experiments. Methane is prescribed as a global concentration value in these experiments. Aerosols percursors are prescribed global emission datasets of black carbon, organic carbon and sulphur dioxide. Ozone precursors are prescribed global emission datasets of $NO_x$, CO, non-$CH_4$ VOCs. Halocarbons are prescribed global concentrations of CFC/HCFC. Climate is represented by prescribed global datasets of sea surface temperatures and sea ice concentrations.

| Scenario | Anthropogenic Air Pollutant Precursors | | Methane | Halocarbons | Climate |
| | Aerosols | Ozone (non-$CH_4$) | | | |
|---|---|---|---|---|---|
| histSST | Transient | Transient | Transient | Transient | Transient |
| piSST | Transient | Transient | Transient | Transient | Fixed at 1850 |
| 1950HC | Transient | Transient | Transient | Fixed at 1950 | Transient |
| piAer | Fixed at 1850 | Transient | Transient | Transient | Transient |
| piCH4 | Transient | Transient | Fixed at 1850 | Transient | Transient |
| piO3 | Transient | Fixed at 1850 | Transient | Transient | Transient |
| piNOx | Transient | $NO_x$ only fixed at 1850 | Transient | Transient | Transient |
| piVOC | Transient | CO and VOC fixed at 1850 | Transient | Transient | Transient |

Results have been obtained from UKESM1-0-LL for all the other transient historical sensitivity experiments, as this was the only model to complete all of the additional experiments requested as part of AerChemMIP (Collins et al., 2017) and thus provide hourly surface ozone concentrations. These additional sensitivity experiments involved running the historical transient simulation but with different short lived climate forcers fixed at pre-industrial values (following an "all but one" methodology), including $CH_4$, $NO_x$, CO, non-$CH_4$ VOCs and aerosols. A separate experiment was also conducted to consider the impact of ozone depleting substances (ODS - chlorofluorocarbons and hydrochlorofluorocarbons) on stratospheric ozone by fixing their concentrations at 1950 values. An additional experiment to those specified in AerChemMIP was conducted to examine the influence of historical climate change, where the underlying climate conditions (sea surface temperatures and sea ice cover) were kept fixed at 1850 values throughout the whole historical time period. A full list of the experiments analysed here is presented in Table 1.

Hourly mean surface ozone concentrations were obtained from each of the experiments for a single model realization over the entire historical period (1850-2014). These values were then used to calculate the relevant peak season ozone health metric (OSDMA8) by first calculating the daily maximum of 8 hour running mean values over a 24 hour period and then a 6 month running mean is calculated from these values. An annual maximum is then calculated to represent the seasonal maximum daily exposure values (avoiding biases from different peak seasons across the world), which is consistent with the metric used in the current WHO air quality guideline values (https://apps.who.int/iris/handle/10665/345329). Regional mean OSDMA8 values have been calculated for the 21 regions used in the GBD study (region definitions shown in Figure A1).

Surface ozone concentrations are simulated at a relatively coarse resolution (>100km) by the CMIP6 models used in this study, which can pose problems when assessing the associated impacts on air quality and human health. Additionally, previ-

ous evaluations have shown that global chemistry climate models tend to overestimate surface ozone concentrations (Young et al., 2018; Turnock et al., 2020). To provide a more accurate representation of surface ozone concentrations, which is necessary when linking concentrations to health effects from long-term exposure, the concentrations simulated by each model in the present day period (2005-2014) have been corrected for biases against the Regionalized Air Quality Model Performance (RAMP) dataset (Becker et al., 2023). The RAMP dataset provides a high-resolution ($0.1°$ x $0.1°$) global surface ozone dataset

from 1990 to 2017 by using state-of-the-art geostatistical methods to integrate surface ozone observations with output from multiple atmospheric chemistry models (including contributions from the GFDL-AM4 model - the atmosphere model version of GFDL-ESM4). The RAMP dataset directly provides ozone concentrations as OSDMA8 values, which are used to correct the corresponding OSDMA8 values simulated by each CMIP6 model. First, each model is regridded to the finer spatial resolution of the RAMP dataset. We then use OSDMA8 values over the last 10 year mean period (2005-2014) from the RAMP dataset to

correct for the absolute difference simulated in the same time period by the CMIP6 models. Using the most recent 10-year mean period of 2005-2014 from the historical experiments maximises the availability of ground-based observations in the RAMP dataset used for correction. Figure 1 shows the 10-year mean (2005 to 2014) simulated OSDMA8 values from each CMIP6 model in the histSST experiment and also the difference compared to the RAMP dataset in the same time period (i.e. the correction applied to each CMIP6 model). EC-Earth3-AerChem shows a consistent overestimation of OSDMA8 concentrations

across all regions (up to 20 ppb), whilst GFDL-ESM4 and UKESM1-0-LL have smaller underestimations and overestimations of OSDMA8 values across different regions. Applying this correction to each model for the 2005 to 2014 10-year mean period provides a present-day climatological baseline for each model that can be used as a baseline for long term changes and also in the calculation of the human health response to exposure, as the simulated concentrations align with observations now as much as possible and the known biases in surface ozone concentrations have been reduced. Staehle et al. (2024) evaluated the

performance of four different techniques for bias correcting surface ozone in global chemistry-climate models. The techniques assessed included: mean bias, relative bias, delta correction and quantile mapping. Staehle et al. (2024) recommended using the delta correction method for correcting future projections of surface ozone due to its lower errors compared to mean and relative bias, and numerical simplicity compared to quantile mapping. Delta correction has also been used as a method in other recent air pollution health studies (Akritidis et al., 2024; Pozzer et al., 2024). Therefore, based on the recommendations of

Staehle et al. (2024) and other recent studies, we use the delta correction method to calculate the change in historical ozone concentrations by applying the ratio of change between each decadal mean and the present day (2005-2014) on top of the bias corrected baseline for each model. This then generates a new historical time series for each experiment that is connected to the observed present day values. The temporal changes in OSDMA8 values (from histSST) are calculated by comparing each 10-year mean period to the 2005-2014 baseline period. The contribution of each individual historical driver is obtained through

a comparison of different pairs of bias corrected model experiments i.e. a control experiment (histSST) with an individual sensitivity experiment where an individual driver is fixed (e.g. $piNO_x$). The contribution of each individual driver is calculated by taking the difference between OSDMA8 concentrations in the same 10-year mean time period between each of these paired experiments. For example, to quantify the contribution of $NO_x$ emissions to OSDMA8 concentrations in the 2005 to 2014 time period, the 10-year mean OSDMA8 values in this time period from the $piNO_x$ experiment (where $NO_x$ emissions are fixed

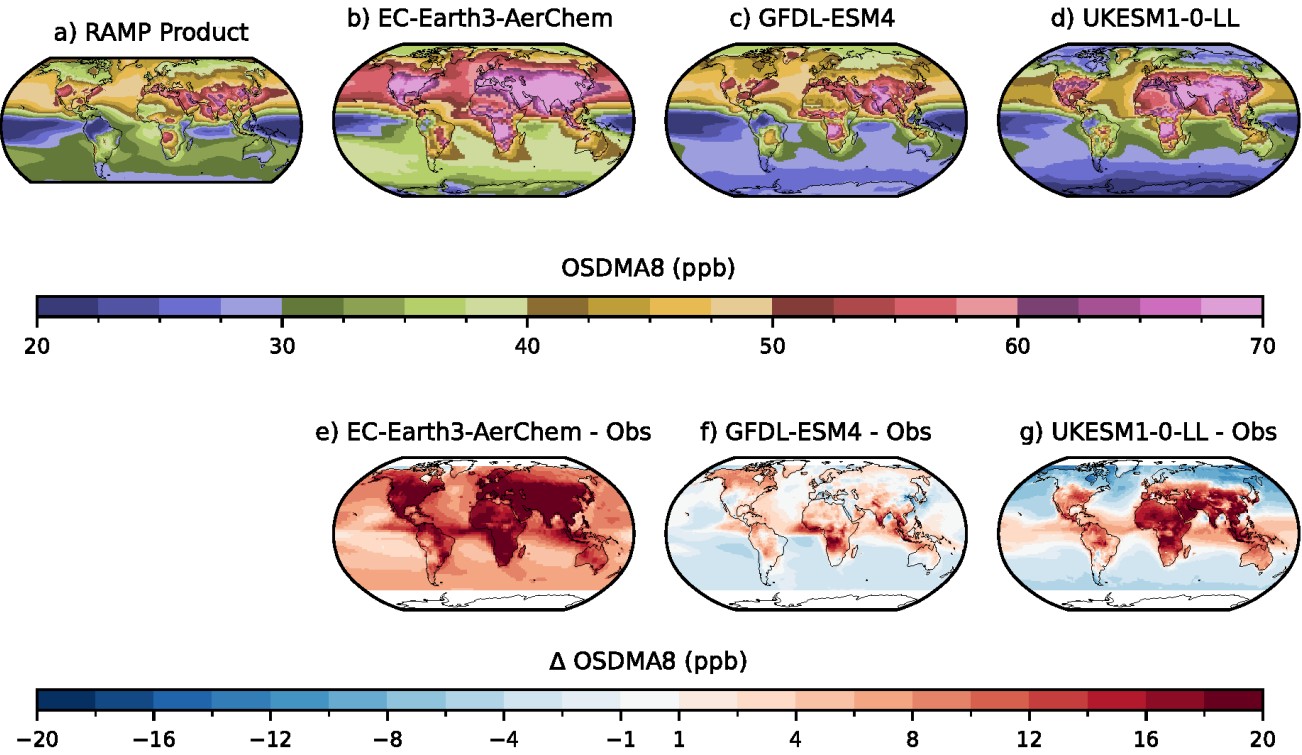

**Figure 1.** 10 year mean surface OSDMA8 values (2005 to 2014) from the a) RAMP observational dataset (Becker et al., 2023) and simulated by 3 CMIP6 models b) EC-Earth3-AerChem, c) GFDL-ESM4 and d) UKESM1-0-LL and difference in the same models e) to g) when compared to the RAMP observational dataset (Becker et al., 2023) over the same time period.

at 1850 values) are subtracted from the values in the same time period of the histSST experiment experiment (where $NO_x$ emissions have changed) i.e. histSST minus $piNO_x$. This example quantifies the change in OSDMA8 concentrations in the 2005 to 2014 time period to $NO_x$ emissions if they had increased from 1850 to 2014 values. This method is then repeated for each individual driver using the relevant control and sensitivity experiment pairs.

A feature of the histSST sensitivity experiments conducted by UKESM1-0-LL is that they use prescribed values of global $CH_4$ concentrations, which are unable to respond to any other perturbation of the inputs (e.g., fixing $NO_x$ emissions at 1850 values), in the same way as a $CH_4$ emission-driven model would. This experimental set up therefore neglects the impacts

of any of the sensitivity experiments on global $CH_4$ concentrations and also consequently ozone, as $CH_4$ is a precursor to ozone formation. Therefore, in each of the histSST sensitivity experiments we have adjusted the ozone concentrations to take account for the consequences of changes in the $CH_4$ lifetime (from the feedback on its own concentrations). Firstly, a global $CH_4$ lifetime is calculated for each experiment, including histSST. The $CH_4$ feedback factor (Prather, 1996) is then calculated by using the difference in the $CH_4$ lifetime and $CH_4$ concentrations in the histSST and histSST-piCH4 experiments over the 30-year period 1930 to 1960, when large changes in $CH_4$ concentrations occur but there are smaller influences from other factors, e.g., halocarbons (Stevenson et al., 2020). The $CH_4$ feedback factor is calculated to be 1.30 over this 30-year period, which is similar to other previous estimates (O'Connor et al., 2021; Stevenson et al., 2013). The feedback factor and $CH_4$ lifetime can then be used to calculate the equilibrium $CH_4$ concentrations in each experiment from the transient prescribed values used. The difference in prescribed and equilibrium $CH_4$ concentrations can be used as input to the relationship between $CH_4$ perturbations and ozone response derived by Wild et al. (2012) and subsequently updated by Turnock et al. (2018). This relationship was obtained by analysing the ozone response of multiple chemistry transport models in an experiment where the global $CH_4$ concentrations was reduced by 20%; conducted as part of phase 1 and 2 of the Hemispheric Transport of Air Pollutant (HTAP) project. A non-linear relationship is then used to scale this ozone response (calculated from the HTAP 20% global $CH_4$ reduction experiment) by the ratio of the global $CH_4$ abundance change in this study (the different between equilibrium and prescribed concentrations) relative to the 20% abundance change used in the original HTAP experiments. From this we now calculate for each sensitivity experiment the response in surface ozone concentrations that occurs due to the change in $CH_4$ concentrations resulting from the adjustment of $CH_4$ lifetime. The model simulated ozone concentrations are then adjusted accordingly in each of the histSST sensitivity experiments for this change. Four of the sensitivity experiments (piSST, 1950HC, piNOx and piO3) result in an increase in $CH_4$ concentrations and subsequently a relatively small increase in ozone. Whereas, in the other three sensitivity experiments (piAer, piCH4 and piVOC) there is a small reduction in $CH_4$ and ozone concentrations. The magnitude of adjustment to the ozone response slightly varies across regions in each sensitivity experiment, although the sign of change is the same. These ozone values adjusted for the impact of $CH_4$ lifetime can then be used to calculate the historical changes in surface ozone values relative to the bias-corrected baseline period.

## 2.2 Emissions in Historical Scenarios

There have been large changes in emissions of ozone precursors (CO, $NO_x$, non-$CH_4$ VOCs and $CH_4$) since 1850 due to increasing human industrial activity and economic development (Hoesly et al., 2018). Figure 2 shows the relative change (compared to 1850 values) in these ozone precursors that are used as input to the CMIP6 historical experiments and sensitivity experiments considered in this study (Table 1). Global anthropogenic emissions of $NO_x$ have seen the largest increase of the ozone precursor emissions, a fractional increase of >10 globally since 1850. Global emissions of non-$CH_4$ VOCs and CO, along with global $CH_4$ concentrations have all increased globally by a fractional change of >1 (i.e. more than doubled) since 1850. These large changes will all have a substantial effect on surface ozone concentrations throughout the historical period considered in the CMIP6 experiments (1850 to 2014). Additionally, changes in climate and stratospheric ozone concentrations

over the historical period will also have an influence on surface ozone concentrations; the changes in these are considered in the sensitivity experiments in Table 1. The transient change in surface air temperature simulated by UKESM1-0-LL over the historical period (Fig. 2 right panel) showed particularly large cold biases compared to observations throughout the latter half of the $20^{th}$ Century (peak of $\approx$0.5k), which was similar to a number of other CMIP6 models (Flynn and Mauritsen, 2020). The cold temperature biases simulated by UKESM1-0-LL have been previously attributed to an excessive aerosol forcing, with recent changes to aerosol and cloud properties in an updated version (UKESM1-1-LL) showing an improved representation of historical surface temperatures (Zhang et al., 2021; Mulcahy et al., 2023). However, in 2014 the global annual mean surface temperature in HadCRUT5 observations (Morice et al., 2021) and UKESM1-0-LL simulations (Fig. 2) both increased by a similar amount of $\approx$1K (compared to a 1850 to 1900 mean period), mainly due to increasing concentrations of long-lived greenhouse gases (Chen et al., 2021). Total column ozone burdens have shown large reductions since the 1960s in UKESM1-0-LL historical simulations due to the reductions in stratospheric ozone from increases in ozone depleting substances, e.g. CFCs. However, total column ozone simulated by UKESM1-0-LL in CMIP6 simulations tends to be higher than other CMIP6 models and observations, and also simulates a stronger depletion of stratospheric ozone from 1960 to 2000 (Keeble et al., 2021). All of these drivers of surface ozone are used as transient changes to the inputs for the histSST experiment and are also fixed at 1850 values (or 1950 for ODS) for the individual sensitivity experiments.

## 2.3  Health Calculations

To assess the health impacts from long-term exposure to ozone concentrations during the historical period, we use the attributable fraction (AF) metric, which is defined here as the percentage of deaths from chronic obstructive pulmonary disease (COPD) attributable to ozone pollution. A full health assessment is not undertaken due to the absence of long term datasets for changes in population (on a latitude longitude grid) and country specific baseline mortality rates over the 1850 to 2014 period. In more detail, AF(x,y,t) is calculated at a given location with x and y coordinates and for a specific time period t, as shown in equation (1):

$$AF(x, y, t) = \frac{RR(x, y, t) - 1}{RR(x, y, t)} \tag{1}$$

where RR(x,y,t) is the relative risk (also known as hazard ratio) for ozone-related excess mortality from COPD. Following the GBD 2019 methodology (Murray et al., 2020), RR is estimated using a log-linear function as shown in equation (2):

$$RR(x, y, t) = e^{\beta(X(x,y,t) - TMREL)} \tag{2}$$

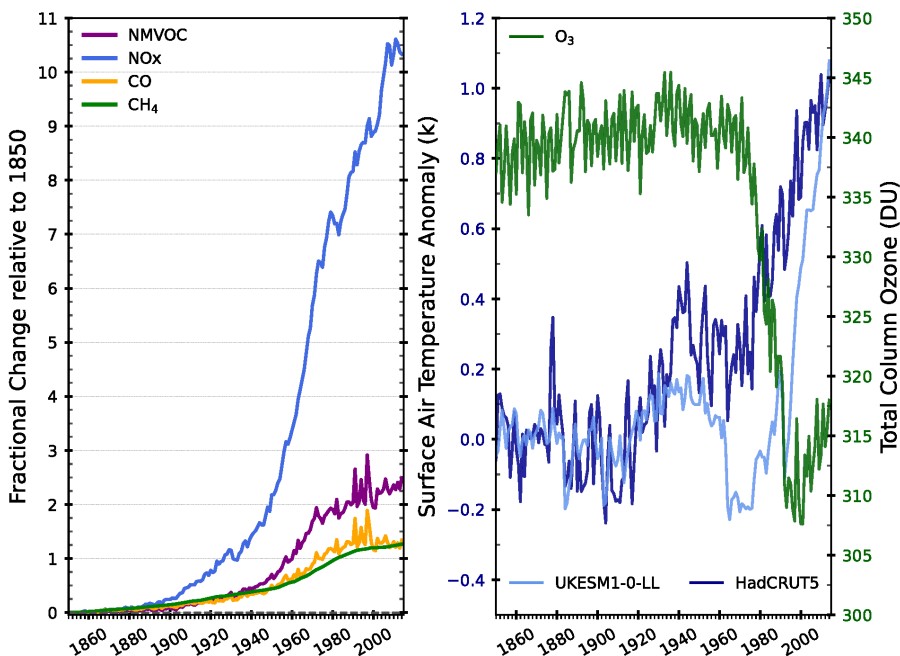

**Figure 2.** Relative change in total historical emissions of ozone precursors ($NO_x$, CO and NM(non-$CH_4$)VOCs) and global $CH_4$ concentrations compared to 1850 values (left panel). Global annual mean surface air temperature anomaly, relative to an 1850-1900 mean period, from UKESM1-0-LL historical simulations and HadCRUT5 (Morice et al., 2021) (blue lines, right panel). Global annual mean total column ozone values from 1850 to 2014 simulated by UKESM1-0-LL (green line, right panel).

where X(x,y,t) is the health-relevant ozone metric (OSDMA8) at the given location and time period and TMREL is the theoretical minimum risk exposure level, below which ozone is considered as not being harmful for human health. For TMREL we adopted the value of 32.4 (29.1–35.7) ppb, while the $\beta$ parameter value(s) result from the RR per 10 ppb of ozone (OSDMA8) value of 1.063 (1.029–1.098), estimated from a meta-regression of five cohorts studies (Murray et al., 2020).

250

Since ozone concentrations (OSDMA8) are the only varying input in the AF calculations, the AF temporal changes over the historical period reflect the changes of the percentage of ozone-related COPD excess mortality from ozone variations only. The ozone exposure values were calculated as 10-year mean OSDMA8 values across the historical period from the bias-corrected model simulation data. An assessment of population exposure to ozone is also provided by calculating the number of people

255 exposed to OSDMA8 concentrations above the TMREL in 3 different decadal time periods (1850-59, 1980-89 and 2005-14). Population data for these periods is obtained for the nearest available years (1850, 1980 and 2010) from the Hyde dataset (History database of the Global Environment) (Klein Goldewijk et al., 2017).

Attribution of ozone health effects (AF) to different drivers by taking the difference between the reference and sensitivity simulations is not valid, as AF is not directly proportional to OSDMA8 concentrations but rather depends on whether ozone

260 exceeds the TMREL and by how much. This type of decomposition would be feasible if the actual percentage contribution of

each driver to the total OSDMA8 change could be estimated, which is not possible in the current approach due to the non-linearities in ozone chemistry. Therefore, only qualitative insights can be made on the relative contributions of each driver to ozone health effects. Thus, AF in a sensitivity simulation should be viewed as a conceptual indicator, reflecting the potential health impacts under a specific scenario (e.g., pre-industrial $NO_x$ emissions in near present-day conditions), without providing a direct attribution of the health effects of ozone to individual drivers in the real-world context.

## 3  Results

### 3.1  Multi-model Historical Changes in Surface Ozone and Risk to Human Health

#### 3.1.1  Multi-model Historical Changes in OSDMA8

The regional mean change in OSDMA8 concentrations over the 1850 to 2014 period from the 3 bias-corrected CMIP6 models is shown in Figure 3. Large changes in OSDMA8 concentrations have been simulated across all regions over the historical period. Globally, the 10-year multi-model mean (+/- 1 S.D. of 3 model values) OSDMA8 concentrations are simulated to have increased by 12 +/- 2.6 ppb (50% increase) between 10-year mean values centered on 1855 and 2010, which is of similar magnitude to annual mean changes simulated by 6 CMIP6 models (Turnock et al., 2020) and also other previous global chemistry climate model intercomparison studies (Young et al., 2013). In addition, this simulated historical change in global OSDMA8 is within the observed change of surface ozone across the Northern Hemisphere of 30 to 70 % (approximately 10 to 16 ppb) from Tarasick et al. (2019). The largest regional mean increases over the same time period have been simulated to occur over South Asia (29.8 +/- 5.3 ppb, 115%), East Asia (24.8 +/- 2.2 ppb, 88%), High-income Asia Pacific (24.8 +/- 0.4 ppb, 110%), North Africa and Middle East (21.4 +/- 4.8 ppb, 78%), Central Europe (21.3 +/- 3.9 ppb, 90%) and High-income North America (18.6 +/- 1.9, 81%). Previously, CMIP6 models were shown to be able to represent the multi-decadal changes in surface ozone concentrations since 1960 at five remote long-term monitoring locations around the world (Griffiths et al., 2021). Even though this comparison is spatially limited, due to the availability of monitoring locations with multi-decadal observational records, it does provide a degree of confidence in the ability of models to simulate long-term changes at specific remote locations, representative of background conditions. Across most regions OSDMA8 concentrations only show small increases up until about 1950. After this time period concentrations rapidly increase (by approximately 50%) to 2005-2014 values, driven by the large rapid changes in all anthropogenic precursors of ozone ($CH_4$, $NO_x$, CO, Non-$CH_4$ VOCs), with the largest relative changes occurring in $NO_x$ emissions (Fig. 2). The increase in concentrations continued across certain regions, most notably South Asia, East Asia and other tropical and sub-tropical regions, due to the continued increases in anthropogenic ozone precursors from socio-economic development across these regions (Figure A3). However, across some regions in the northern mid-latitudes (high-income North America, Europe and high-income Asia Pacific) OSDMA8 concentrations stopped increasing from about the 1980s, remaining at and near these values until the end of the CMIP6 historical period when a slight reduction in concentrations occurred, in agreement with the observations in the RAMP dataset (Fig. 3). The recent change in OSDMA8 concentrations is consistent with the mitigation measures adopted over these regions to improve regional air quality

by reducing primary air pollutant emissions (United Nations Economic Commission for Europe, 2004; EMEP Steering Body and Working Group on Effects of the Convention on Long-Range Transboundary Air Pollution, 2016), shown by a reduction in $NO_x$ emissions from the peak in the 1980-1990s (Figure A3).

Fig. 3 shows that bias correcting the model data to the RAMP dataset has resulted in the simulated OSDMA8 concentrations from all 3 models being similar in the 2005 to 2014 10-year mean period. However, there is still a diversity of 10 to 15% in the model simulated OSDMA8 concentrations in the pre-industrial period (1850 to 1860), with some of the largest model differences (20%) occurring over South Asia. UKESM1-0-LL consistently simulates larger bias-corrected OSDMA8 concentrations across most regions in the first 10 year mean period of 1855, with GFDL-ESM4 tending to have the lowest concentrations of the 3 models. This diversity between models can be attributed to differences in how they represent chemical processes (see Table A1), particularly relevant for the pre-industrial period are differences in the representation of meteorology/climate, as well as the interactive/natural components such as emissions from biogenic sources (BVOCs), lightning $NO_x$ and wetland $CH_4$ (Rowlinson et al., 2020; Wild et al., 2020) In addition, differences in resolution (both horizontal and vertical) and chemical mechanisms between each of the models can also contribute to the highlighted diversity in simulated ozone concentrations (Wild and Prather, 2006; Wild et al., 2020). These structural differences between models also impacts the chemical sensitivity of ozone formation in each model, resulting in the different simulated historical changes in OSDMA8 across the models (Turnock et al., 2020). Given its higher 1855 concentrations, UKESM1-0-LL is the model that tends to simulate the smallest historical change in OSDMA8 values over the historical period (9.2 ppb globally), whilst GFDL-ESM4 simulates the largest (14.3 ppb globally). If the models are not corrected against the observation based dataset then the simulated concentrations show on average a larger 20% variability between models (up to 30% across South Asia), with the largest diversity in concentrations in the 1855 period. The comparison to the RAMP dataset in the 2005 to 2014 period shows that simulated OSDMA8 values tend to be consistently overestimated by all 3 models (Fig. 1, Figure A2). Overestimating OSDMA8 concentrations has important consequences for the assessment of the long-term impact on human health due to the use of theoretical minimum risk exposure level, below which no risk to health is considered (Section 2.3). Using the uncorrected simulated OSDMA8 concentrations from the models could result in an overestimation of the impacts to human health from long-term exposure, and thus the bias-corrected model simulated concentrations are used for any assessment of health impacts.

### 3.1.2 Historical Changes in Ozone exposure and risk to Human Health

The increase in OSDMA8 concentrations over the historical period across all regions has increased the risk to human health via enhanced long-term exposure to ozone. Figure 3 shows how the simulated regional OSDMA8 concentrations compare with the TMREL of 32.4 ppb. All three models simulate concentrations in first 10-year mean period of 1855 that are below this threshold across most regions of the world, resulting in <20% of the world's population in 1850 being exposed to concentrations above this value (Fig. 4a). However, the simulated increase in OSDMA8 concentrations over the historical period has resulted in concentrations being above the TMREL value in almost every region of the world in the near present-day period. This is shown by

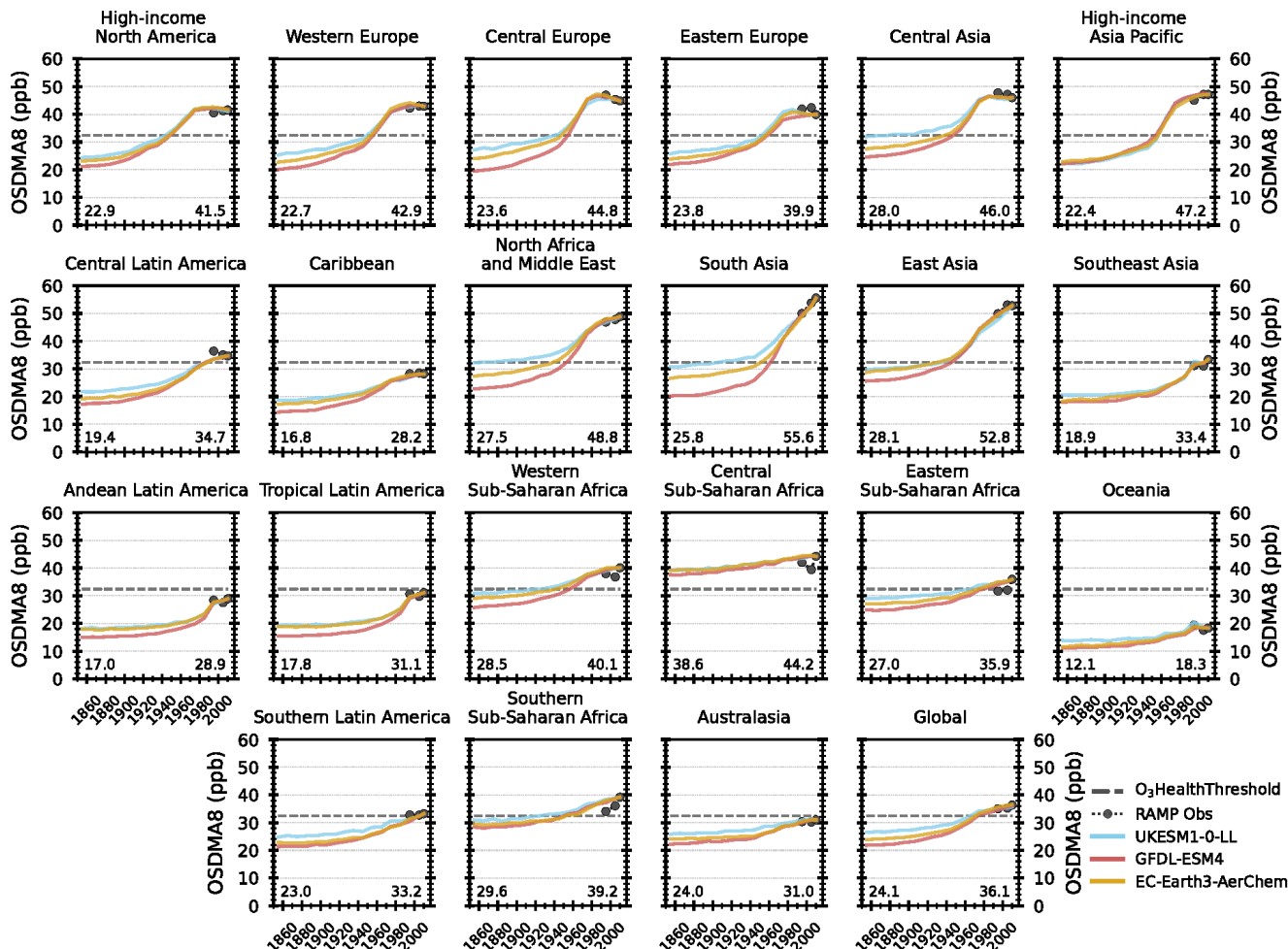

**Figure 3.** Regional mean surface OSDMA8 values from 3 CMIP6 models simulated over the historical period (1850-2014) bias-corrected to a 10-year mean of observations from Becker et al. (2023). Observations from Becker et al. (2023) are shown as circles on each relevant region with the TMREL value of 32.4 ppb as the dashed line. Regional multi-model mean OSDMA8 values are shown for the start and end of the time period.

>90% of the world's population in 2010 being exposed to ozone concentrations above this value, meaning that over the historical period there has been a large increase in the risk to human health from long-term exposure to elevated ozone concentrations.

330    Figure 5 shows the spatial distribution of the multi-model mean attributable fraction over the pre-industrial period 1850-1859 and the near present-day period 2005-2014. High AF values up to ≈ 20% are found over regions of North India, East China, Middle East, and Western United States, indicating that across these regions approximately one out of five COPD deaths in the near present-day period are attributed to long-term ozone exposure.High AF values of >8% also occur over Greenland in 2005-2014, which can be attributed to the presence of the high-altitude ice sheet (≈ 2000m above sea level), meaning that the ozone

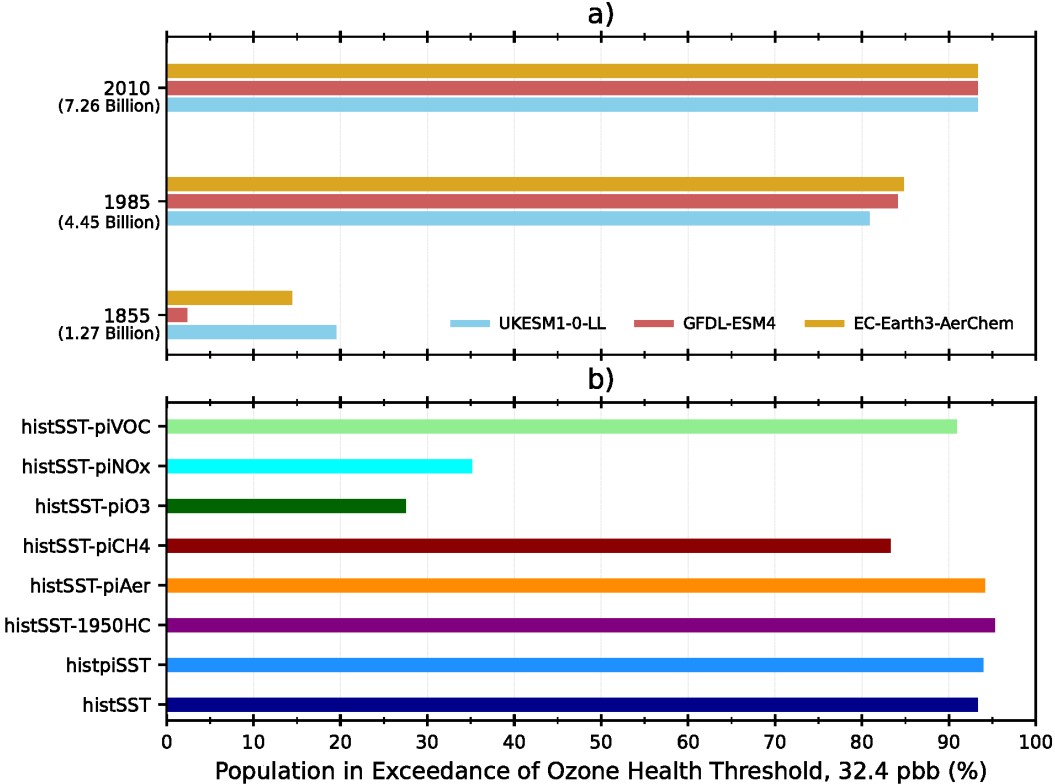

**Figure 4.** Percentage of total global population in exceedance of the theoretical minimum risk exposure level (TMREL - 32.4 ppb) to ozone across 3 historical time periods from 3 different CMIP6 models a) and in the different present day (2010) for sensitivity scenarios with different drivers of ozone fixed b). Total global population for each time period is shown in parenthesis on a).

concentrations are not representative of typical surface values for this latitude but can be considered more like elevated free-tropospheric ozone concentrations. As the AF calculation does not consider changes in population and baseline mortality rates, a distinct increase of the AF is depicted almost globally compared to the 1850-1859 period due entirely to increases in ozone concentrations. As ozone concentrations in the pre-industrial period are below the TMREL value almost all over the world, the respective AF values are near zero. The major exception to this is that AF values are approximately 5% in the 1850-59 period over the Central Sub-Saharan Africa region. This non-zero AF is due to the elevated pre-industrial OSDMA8 concentrations (Fig. 3) arising mainly from biomass burning sources of ozone precursors over this region in this period, although the low population density in this region and time period would represent a low risk to health. Overall This suggests that there is no risk for human health from ozone exposure in the pre-industrial period, although the increases in ozone concentrations up to the near present-day became a major threat for human health, being responsible for more than 10% of COPD deaths in many regions of the Northern Hemisphere.

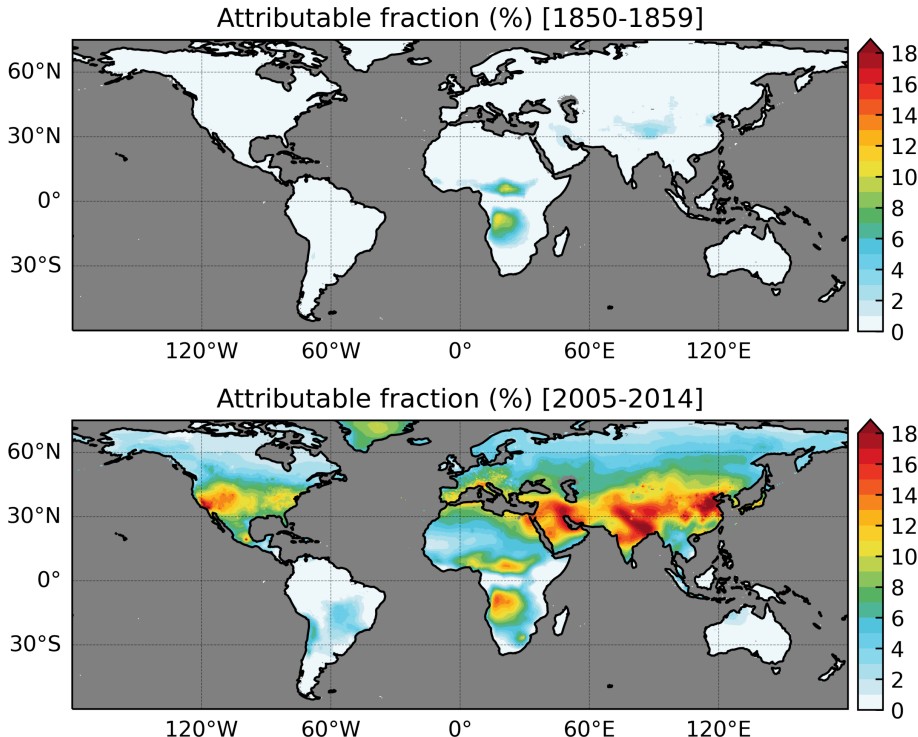

**Figure 5.** Attributable fraction as the percentage of deaths from COPD attributable to ozone pollution calculated from the multi-model mean OSDMA8 values in the 10 year mean periods of 1850-1859 (top) and 2005-2014 (bottom).

## 3.2 Drivers of Historical Change in Surface Ozone and Risk to Human Health

### 3.2.1 Drivers of Historical Change in OSDMA8

Figure 6 shows the change in OSDMA8 concentrations in the present day time period (2005 to 2014) due to the historical changes in all drivers together (histSST) and due to the individual drivers of ozone formation i.e. histSST minus sensitivity experiment. The overall change in global mean OSDMA8 concentrations simulated by UKESM1-0-LL over the historical period in histSST is 9.2 ppb. Considering only historical changes in emissions of ozone precursors (i.e., histSST minus histSST-piO3 = $NO_x$, CO and non-$CH_4$ VOCs) has resulted in a large increase in global mean OSDMA8 concentrations of 9.6 ppb (37%) in the 2005 to 2014 mean period. Of these three precursors, the majority of this change in global mean OSDMA8 concentrations in 2005-14 can be attributed to historical changes in anthropogenic $NO_x$ emissions (8.6 ppb, 32%), compared to CO and non-$CH_4$ VOCs (1.5 ppb, 4%). Regionally, the largest changes in OSDMA8 concentrations occur mainly over the polluted continental regions of the northern mid-latitudes. OSDMA8 concentrations increased by ≈25 ppb over the High-income Asia Pacific region, due to this region having the largest relative increase in anthropogenic $NO_x$ emissions over the historical period (Figure A3). Similarly large increases in OSDMA8 concentrations occur over Europe (16 ppb), High-income North America (16 ppb), East Asia (23 ppb) and South Asia (24 ppb), all mainly due to the large historical changes in anthropogenic $NO_x$

emissions over these regions (Figure A3). Changes in anthropogenic emissions of ozone precursors, particularly over recent decades have also been shown, as in this study, to be the most important driver behind changes in regional surface ozone concentrations (Parrish et al., 2014; Zhang et al., 2016; Lin et al., 2017; Yan et al., 2018; Wang et al., 2022).

Historically, global $CH_4$ concentrations, the other major precursor gas to ozone formation, have more than doubled since 1850 (Fig. 2), which has increased global mean OSDMA8 concentrations by 5.9 ppb (20%) in the 2005 to 2014 period. The impact of changes in $CH_4$ on surface ozone concentrations tends to be larger, in emissions and abundance, than other non-methane VOCs and more globally uniform due to its larger change over the historical period, the longer chemical lifetime of $CH_4$, its larger abundance in the atmosphere, and that it is input to these model experiments as a global annual mean concentration, rather than gridded emissions. This means that the impact of changes in $CH_4$ on surface ozone in this study are spatially limited by the absence of a hemispheric gradient in $CH_4$ concentrations (higher in the northern hemisphere and lower in the southern hemisphere), something that would be provided if using a model with a fully interactive $CH_4$ cycle, including $CH_4$ emissions instead of concentrations (Folberth et al., 2022). The largest increase in OSDMA8 concentrations occurred over parts of Asia and North Africa and Middle East (up to 8.5 ppb, 21%), whilst the smallest increase (2 to 3 ppb, 10 to 13%) occurred over tropical regions of the southern hemisphere. Over some of these more remote (low $NO_x$) regions like sub-Saharan Africa and southern Latin America, historical changes in $CH_4$ had a larger proportional impact on OSDMA8 concentrations, contributing almost as much as changes in $NO_x$ emissions. This shows that historical changes in $CH_4$ concentrations have been important in altering the background ozone concentrations in most regions of the world.

The other drivers of surface ozone formation considered here in the sensitivity experiments are historical changes in climate, aerosols and stratospheric ozone via ODS, which have all tended to have smaller impacts than anthropogenic ozone precursor emissions (Fig. 6 and Figure A4 for more detail). The increasing emissions of ODS since the 1950s reduced stratospheric ozone concentrations (shown as total ozone column on Fig. 2) significantly by the start of the $21^{st}$ Century, which has also led to reductions in OSDMA8 concentrations, potentially via less downward transport to the surface or from changes to photolysis rates. Global mean OSDMA8 concentrations reduced by -2.8 ppb (-7%) in the 10-year mean period of 2005 to 2014 due to smaller amounts of stratospheric ozone from the increased ODS in the histSST experiment compared to the 1950HC experiment in this most recent period. Some of the largest impacts (>-2 ppb) occurred over remote southern hemisphere regions, including southern Latin America and Australasia (Figure A4). In addition, changes in stratospheric ozone also reduced OSDMA8 concentrations across the well known hot spots regions of stratosphere-to-troposphere transport (STT) (Škerlak et al., 2014; Akritidis et al., 2021) of high altitude Central Asia, the Eastern Mediterranean and Middle East, and the Western United States.

Increasing emissions of aerosols and aerosol precursors over the historical period (Figure A5) has resulted in a small reduction of global mean OSDMA8 concentrations of -0.8 ppb (-2%) in the 2005 to 2014 period (see Figure A4 for more detail). Increasing aerosols over the historical period implies that there is an increase in the heterogeneous loss of nitrogen oxides to

aerosol surfaces (the only active heterogeneous tropospheric mechanism currently in UKESM1-0-LL (Archibald et al., 2020b)), leading to a reduction in ozone formation. This effect is probably underestimated due to the absence of the loss mechanism involving $HO_2$ uptake on aerosols in UKESM1-0-LL (Ivatt et al., 2022). Regionally, increasing aerosols causes the largest reduction in OSDMA8 concentrations of up to -2.1 ppb (-4%) across Asia (central, south and east), North Africa and Middle East regions in the 2005 to 2014 period when aerosol concentrations were higher. However, the largest reduction on OSDMA8 concentrations from aerosols (-2.7 ppb, -6%) occurred in the 1980s across Europe. Aerosol concentrations peaked in Europe at this time due to the subsequent enacting of air pollutant controls, meaning aerosol concentrations and their impact on surface ozone were reduced thereafter (Turnock et al., 2015).

The climate change signal over the historical period has resulted in an approximate 1K increase in global mean surface temperatures simulated by UKESM1-0-LL by the 2005 to 2014 period (Fig. 2), which has resulted in a small reduction in global mean OSDMA8 concentrations of -0.8 ppb (-2%) that is consistent with the surface ozone temperature sensitivity of -0.79 ppbv $^{\circ}C^{-1}$ from Zanis et al. (2022). The global mean reduction in ozone has mainly been driven by decreases over the ocean, which can be attributed to more ozone destruction here due to increased hydroxyl formation from enhanced amounts of water vapour occurring in a warmer world (Johnson et al., 1999; Doherty et al., 2013; Zanis et al., 2022). Historical climate change has resulted in enhanced ozone formation rates over some polluted continental regions including Southeast Asia, Central Europe, east Asia, high-income Asia Pacific and parts of North America (see Figure A4 for more detail), which is shown by a small increase of OSDMA8 concentrations (<1 ppb, 2%). There has also been a similar small increase in OSDMA8 concentrations across some of major biogenic emission regions (Central Sub-Saharan Africa and parts South America). This suggests a small climate impact on natural sources of ozone precursor emissions is simulated by the interactive BVOC emission scheme included within UKESM1-0-LL. The magnitude of the changes in ozone due to 1K of warming over the historical period are consistent with studies analysing the change of ozone in response to future warming (Archibald et al., 2020c; Zanis et al., 2022), with the impact of climate change on surface ozone being less in this study due to the smaller level of historical warming compared to those projected for the future warming of 2.7 to 5.4K. In addition, Zanis et al. (2022) showed that UKESM1-0-LL had a lower ozone sensitivity per unit temperature change on a global mean basis (-0.79 ppbv $^{\circ}C^{-1}$) than other CMIP6 models (multi-model mean of -0.93 ppbv $^{\circ}C^{-1}$), which could mean that climate change impacts on surface ozone are underestimated in this study. However, this global sensitivity number obscures the spatial variation in the sensitivity of surface ozone to temperature (both positive and negative) across different models (Zanis et al., 2022). The model diversity highlights that the surface ozone response to climate change is still uncertain in global chemistry climate models and that both a stronger and weaker response is possible across different regions.

In summary, the model experiments isolating the impact of the individual drivers shows that historical changes in anthropogenic $NO_x$ emissions contribute the most, approximately 42%, to the summed total change in historical global mean OSDMA8 concentrations from all drivers combined. The contribution to historical changes in global mean OSDMA8 concentrations from other drivers was 29% from global $CH_4$ concentrations, 14% from stratospheric ozone, 7% from anthropogenic

emissions of CO and non-$CH_4$ VOCs, and 4% from both aerosols and climate change. This attribution of drivers is in agreement with other studies that showed the reductions in peak season surface ozone over North America and Europe and increases over east and south Asia between 1980 and 2010 were primarily driven by changes in anthropogenic ozone precursor emissions, with much smaller contribution from other factors including changes in $CH_4$ concentrations and climate (Zhang et al., 2016; Yan et al., 2018; Wang et al., 2022). However, the linear combination of the historical change in global mean OSDMA8

concentrations, calculated from the individual driver experiments by UKESM1-0-LL (+11.6 ppb), is found to be about 20% larger than that calculated when all drivers are varied simultaneously (+9.2 ppb from histSST). This indicates at a global scale that there are feedbacks and interactions (e.g., clouds or aerosols influencing photolysis rates) influencing ozone formation in each of the individual experiments due to the changes from that particular driver (Gao et al., 2020; Qu et al., 2021). These feedbacks/interactions then have a non-linear effect on the formation of ozone, reducing the magnitude of change in OSDMA8

at the global scale when the drivers are changed in combination in single experiment (histSST). At the global scale, the impact on ozone concentrations due to each individual driver can be considered overestimated as change in the combined driver experiment (histSST) is smaller than those from the linear addition of the individual driver experiments.

### 3.2.2 Drivers of change in Ozone Exposure and risk to Human Health

The change in the individual drivers of ozone formation show different impacts of OSDMA8 concentrations over the historical period, which will also have an impact on the risk to human health via changes to long-term exposure. Figure 4b) shows a comparison of the different fraction of the global population in 2010 that are exposed to OSDMA8 concentrations above the TMREL of 32.4 ppb in each of the sensitivity scenarios that isolate the different drivers of ozone formation. If emissions of ozone precursors ($NO_x$, CO and non-$CH_4$ VOCs) are kept fixed at 1850 values then the fraction of the near present-day global

population exposed to OSDMA8 concentrations above 32.4 ppb is reduced substantially to 27%, from >90% when emissions are allowed to increase in histSST. Fixing solely anthropogenic $NO_x$ emissions at 1850 is shown to be the dominant cause of this reduction by decreasing the global population exposure above the TMREL to 34%. Fixing the emissions of the other precursors (CO and non-$CH_4$ VOCs) at 1850 values results in a very small (2%) change in global population exposure. A relatively small impact on global population exposure is shown when fixing global $CH_4$ concentrations at 1850, which reduces

the proportion of the global population exposed to OSDMA8 concentrations above TMREL to 83% (from >90%). Fixing the other historical drivers of ozone concentrations (climate change, stratospheric ozone and aerosols) slightly increases the global population exposure to OSDMA concentrations above the TMREL by 1 to 2%. These results show that historical changes in anthropogenic $NO_x$ emissions are the leading cause of increased ozone exposure that is considered harmful to human health.

Figure 7 presents the global and regional mean AFs for the histSST and sensitivity experiments isolating the different drivers of ozone formation in the near present-day period 2015-2014. A high health risk from COPD attributable to ozone pollution (histSST) is found in South Asia and East Asia with a regional average AF of 13% and 12%, respectively; while values in other regions are: North Africa and Middle East (9%), High-income Asia Pacific (9%), Central Europe (7%), Western Europe

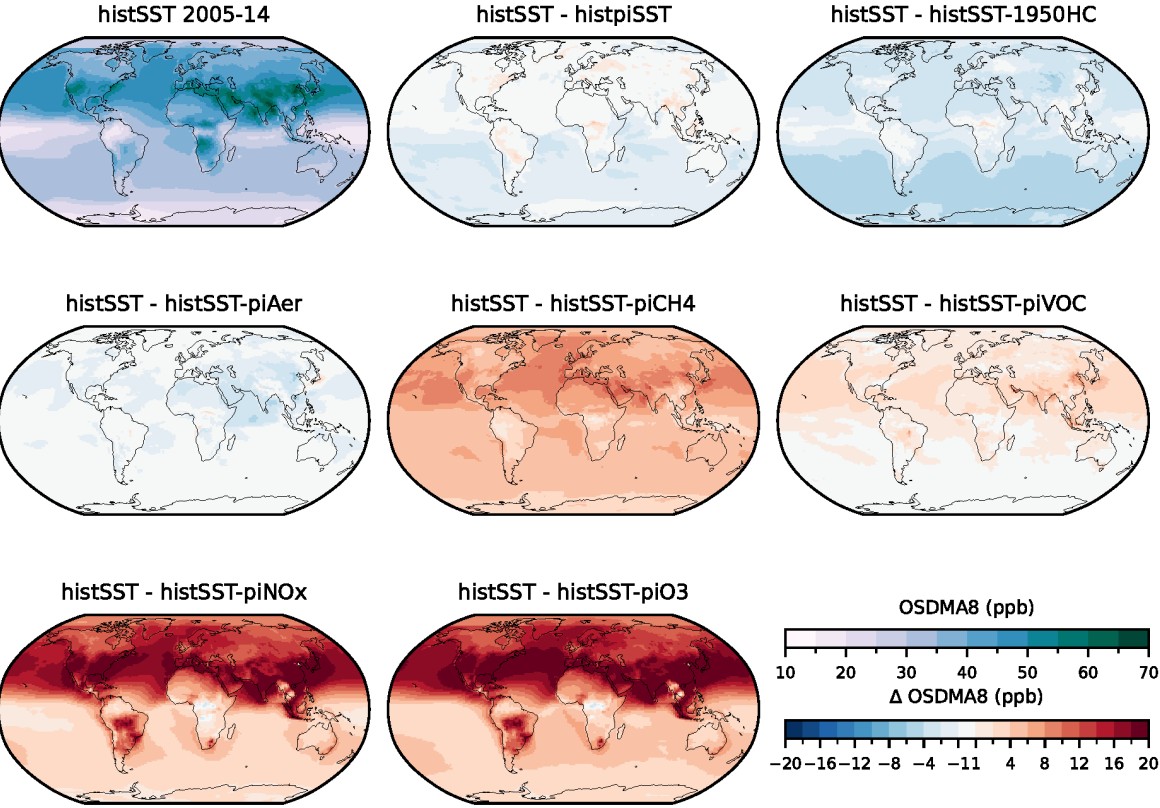

**Figure 6.** 10-year mean in surface OSDMA8 values simulated by UKESM1-0-LL in the HistSST experiment for 2005-14 (bias-corrected) and the change in the same time period in the HistSST experiment relative to each of the sensitivity experiments with different drivers of ozone fixed

(6%) and High-income North America (5%). Regarding the drivers of ozone changes, fixing NOx emissions to pre-industrial

levels (histSST-piNOx) eliminates the risk for human health from ozone exposure for the 2005-2014 period in most regions. Historical increases in methane concentrations also appear to imply a risk for human health in near present-day both globally and regionally, as setting methane concentrations to pre-industrial levels results in a lower risk for health in the highly populated regions of South Asia, East Asia, and Western Europe with AF values of 9%, 7%, and 2%, respectively. Historical changes in climate, aerosol precursor emissions and stratospheric ozone have relatively small impacts on historical OSDMA8 changes

and consequently on regional mean near present-day AF. Isolating these drivers at 1850 levels (or 1950 for stratospheric ozone) shows only small changes in AF values and therefore in ozone-related health risk.

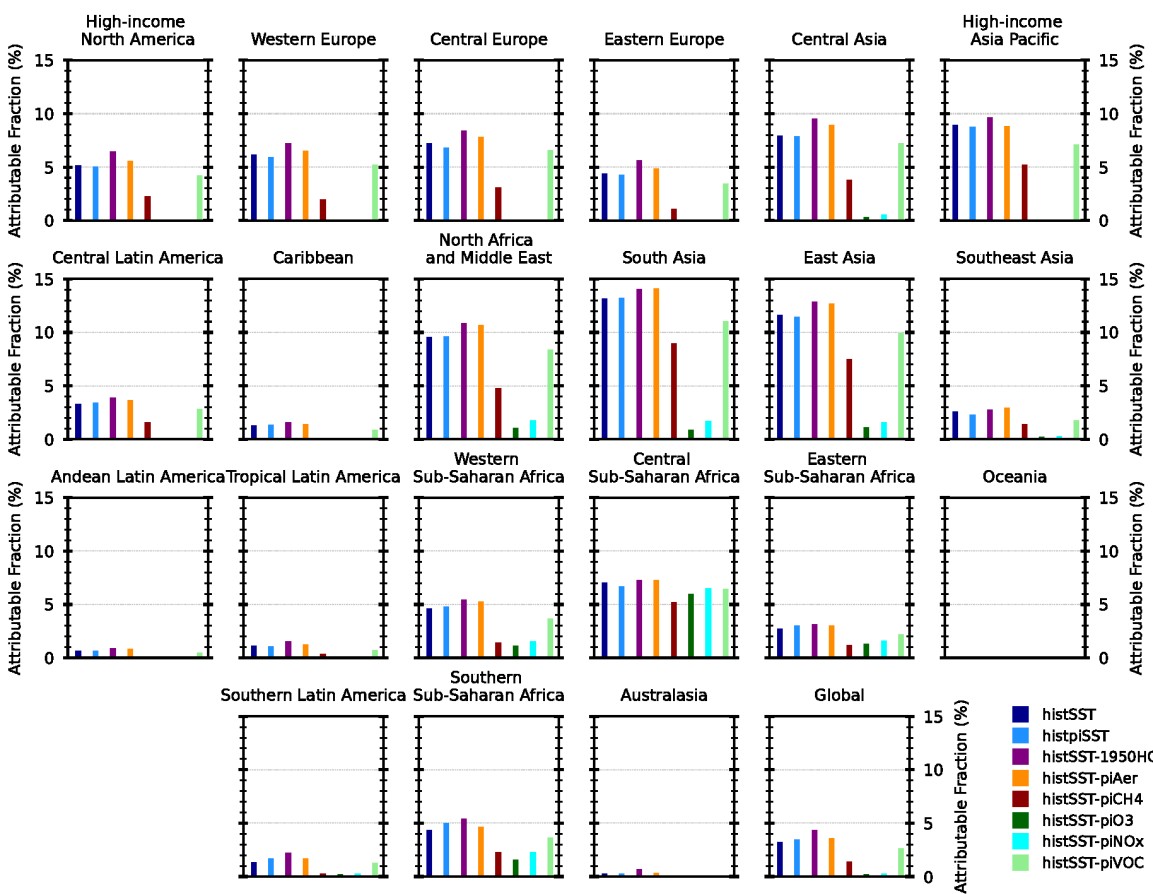

**Figure 7.** Regional mean attributable fraction as the percentage of deaths from COPD calculated from the 10 year (2005-14) mean OSDMA8 values simulated by UKESM1-0-LL in the HistSST and sensitivity experiments with different drivers of ozone fixed

## 4 Conclusions

Surface ozone is an important greenhouse gas and secondary pollutant in the lower atmosphere, influencing both the Earth's climate and air quality. Exposure to elevated concentrations of ozone at the surface can have detrimental impacts on human
health. In this work we use numerical results of hourly surface ozone, obtained from the latest generation of global chemistry climate models that participated in CMIP6, as part of the Aerosol and Chemistry Model Intercomparison Project (AerChem-MIP). For the first time we use hourly surface ozone output from three different CMIP6 models, over the 165-year historical period (1850 to 2014), to calculate changes in the peak season ozone metric (OSDMA8), that can relate risks to human health from long-term exposure surface ozone. All three CMIP6 models tended to overestimate the recent observed OSDMA8 concen-
trations, in agreement with other recent multi-model evaluations of surface ozone concentrations (Young et al., 2018; Turnock et al., 2020). The most recent 10-year mean period (2005 to 2014) of OSDMA8 concentrations simulated by the three CMIP6 models were bias-corrected to the observations to avoid overestimating the present-day risk to human health from long-term

ozone exposure. However, there is still a 10 to 15% difference in the simulated OSDMA8 concentrations in the pre-industrial period (1850 to 1859) due to physical and chemical differences across the three models, which also influences the magnitude of the simulated historical change in OSDMA8. Global increases of more than 50% in OSDMA8 concentrations were simulated over the historical period by all three CMIP6 models, with the largest changes over the Northern Hemisphere regions (North America, Europe, Asia). This change has increased the proportion of the global population being exposed to OSDMA8 concentrations above the theoretical minimum risk exposure level (TMREL), below which ozone is considered as not being harmful for human health, from less than 20% in 1855 to more than 90% in 2010. The risk to human health from long-term exposure to ozone concentrations has been assessed in terms of changes to the attributable fraction (AF), defined here as the percentage of deaths from chronic obstructive pulmonary disease (COPD) attributable to ozone pollution. The AF has increased by more than 10% over many regions of the Northern Hemisphere, showing that increased ozone concentrations over the historical period have contributed to a significant increase in the long-term risk to human health across many populated regions of the world. In addition, the historical increase in surface ozone concentrations has made the risk of human mortality from long-term exposure to ozone similar to other environmental risk factors, but still much less than from long-term exposure to fine particulate matter (GBD 2021 Risk Factors Collaborators, 2024).

An analysis of the drivers of historical changes in OSDMA8 concentrations and risks to human health is provided from hourly surface ozone outputs of sensitivity experiments conducted using a single CMIP6 model that isolate the historical changes in ozone precursors ($NO_x$, CO, non-$CH_4$ VOCs and $CH_4$), anthropogenic aerosols, ozone depleting substances and climate change. Historical increases in anthropogenic emissions of $NO_x$ is identified as the leading contributor to increases in OSDMA8 concentrations ($\approx 40\%$ globally) over the period 1850 to 2014, particularly over Northern Hemisphere regions. Increases in global $CH_4$ concentrations is also shown to have an important contribution to the historical changes in OSDMA8 concentrations ($\approx 30\%$ globally), particularly over more southern hemisphere regions where its contribution can be as large of that from $NO_x$. The increase in anthropogenic emissions of other ozone precursors (CO and non-$CH_4$ VOCs) has had a relatively small impact on increasing historical OSDMA8 concentrations ($\approx 7\%$ globally) compared to $NO_x$ and $CH_4$. The other drivers considered here (stratospheric ozone, aerosols and climate change) all have relatively small contributions to historical changes in OSDMA8 concentrations. The depletion of stratospheric ozone since the 1950s has resulted in less downward transport of ozone to the surface and a small reduction in historical surface concentrations, particularly over more remote Southern Hemisphere regions. Increasing anthropogenic emissions of aerosols also caused a small reduction in OSDMA8 concentrations, particularly over Asia, although this effect might be underestimated due an under representation of heterogeneous chemistry mechanisms within the model (Ivatt et al., 2022). Historical climate change contributed to a small reductions in historical OSDMA8 concentrations over remote regions and a small increase over polluted continental regions due to changes in ozone production and destruction pathways. The response of ozone to climate change was small due to the small 1K of global warming considered over the historical period, although the results were consistent with projected changes due to future warming (Zanis et al., 2022).

Increasing anthropogenic $NO_x$ emissions over the 1850 to 2014 period is shown to be the main contributor to the long-term health effects from ozone exposure in the near present day period (2010). Fixing anthropogenic $NO_x$ emissions at 1850 reduces the proportion of the near present day global population exposed to OSDMA8 concentrations above the TMREL to 34%, from more than 90% when they are allowed to increase over the historical period along with other precursors. Fixing $NO_x$ emissions at 1850 also reduces the AF to near zero across most regions, thereby eliminating the risk to human health from long-term ozone exposure. Historical global $CH_4$ concentrations are shown to be the next most important contributor to the health effects from long-term ozone exposure. Fixing global $CH_4$ concentrations at 1850 values reduces the proportion of the global population exposed to OSDMA8 concentrations above the TMREL to 83%, and reduces the AF, and risk to human health, by about half in populated regions. However, the influence of the other drivers is small, having little impact on the health risk of the near present day population from exposure to elevated surface ozone concentrations.

The results from these experiments provide a unique opportunity to quantify the long term changes, and attribution to multiple drivers behind the changes, of an ozone exposure metric relevant to health impacts over a 165-year period from 1850 to 2014. Substantial increases have occurred in surface ozone concentrations and the risk to human health from long-term exposure to these elevated concentrations over the historical period. These changes can be mainly attributed to changes in anthropogenic sources of ozone precursors such as $NO_x$ and $CH_4$, with smaller contributions from CO and non-$CH_4$ VOCs. Sensitivity studies show that drastic changes to these main drivers can have large impacts on the risk to human health. However, there are also non-negligible changes due to other drivers of ozone (stratospheric ozone, aerosols and climate change), which could become more significant in the future depending on the particular emissions and climate pathway the world takes. Therefore, understanding the historical changes and drivers of ozone and human health could help to inform future policy measures.

*Data availability.* The hourly surface ozone data used in this study has been obtained from the CMIP6 data archive which is hosted at the Earth System Grid Federation and is freely available to download from https://esgf-node.llnl.gov/search/cmip6/. The references for AerChemMIP data used from GFDL-ESM4, EC-Earth3-AerChem and UKESM1-0-ll are Horowitz et al. (2018); EC-Earth Consortium (2020); O'Connor (2019). The CMIP6 variable name sfo3 has been used from the experiments listed in Table 1. Processed model data and calculations related to human health that are used in this study can be found on zenodo at https://doi.org/10.5281/zenodo.13385648. The RAMP surface ozone observational data is from Becker et al. (2023).

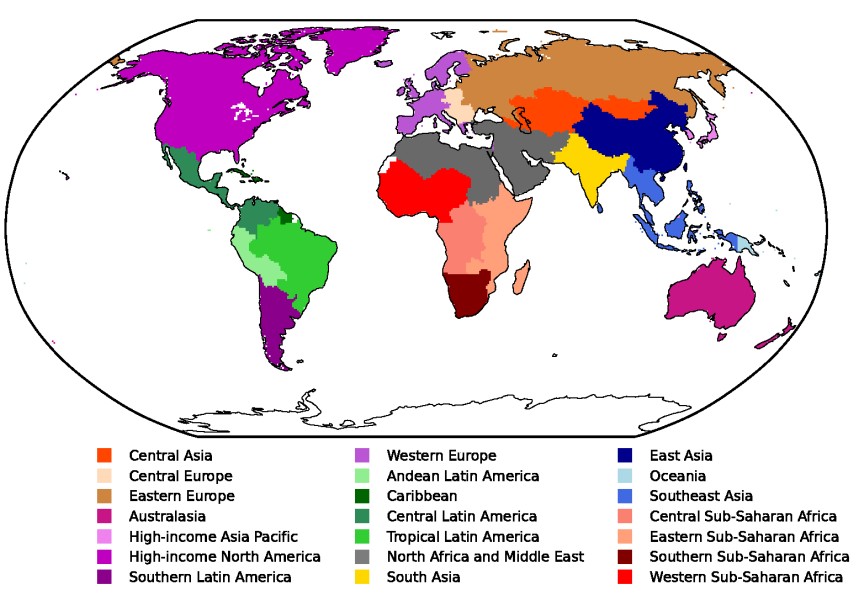

**Figure A1.** 21 regions used in this study based on those from the Global Burden of Disease study

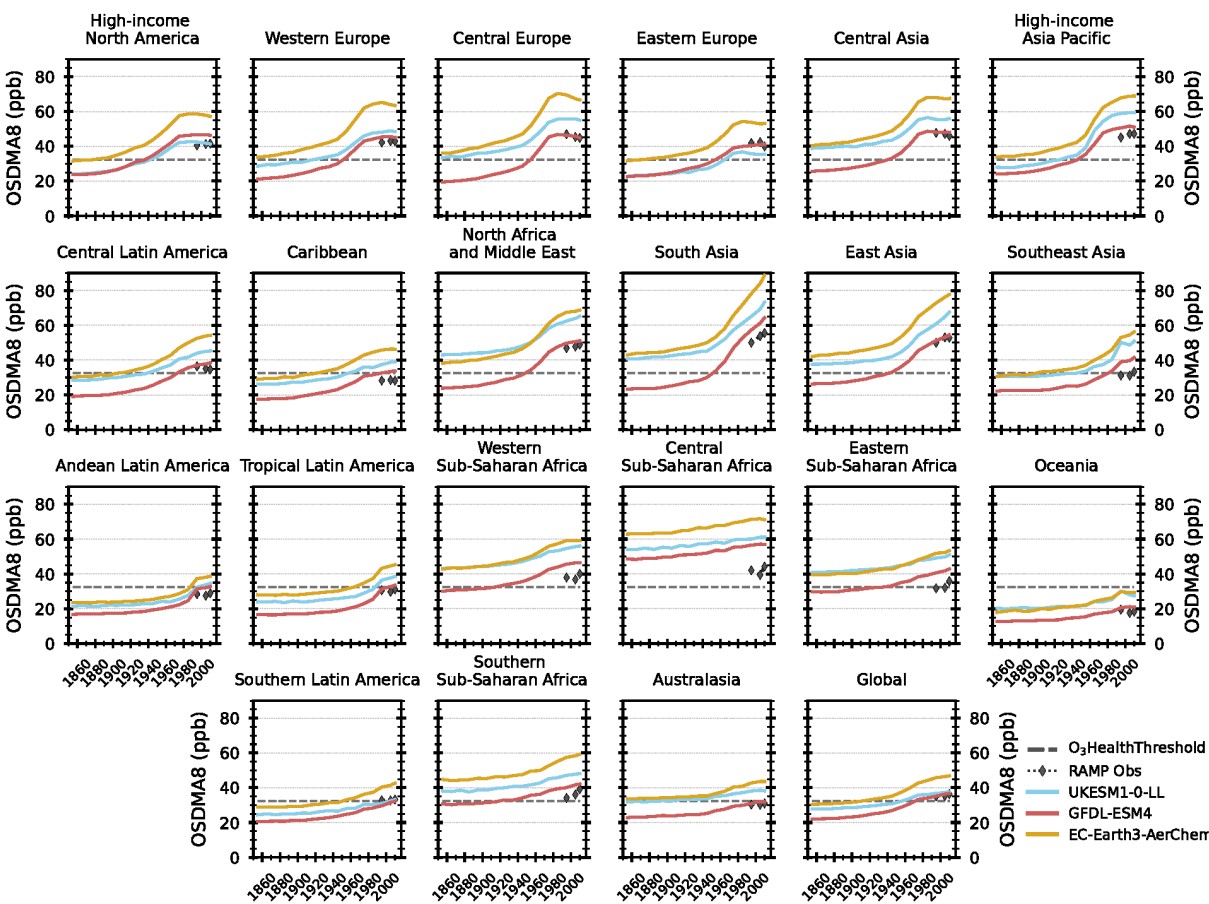

**Figure A2.** Uncorrected 10-year mean regional mean surface OSDMA8 values from 3 CMIP6 models simulated over the historical period (1850-2014) with 10-year mean values from RAMP observations (Becker et al., 2023) shown in grey diamonds.

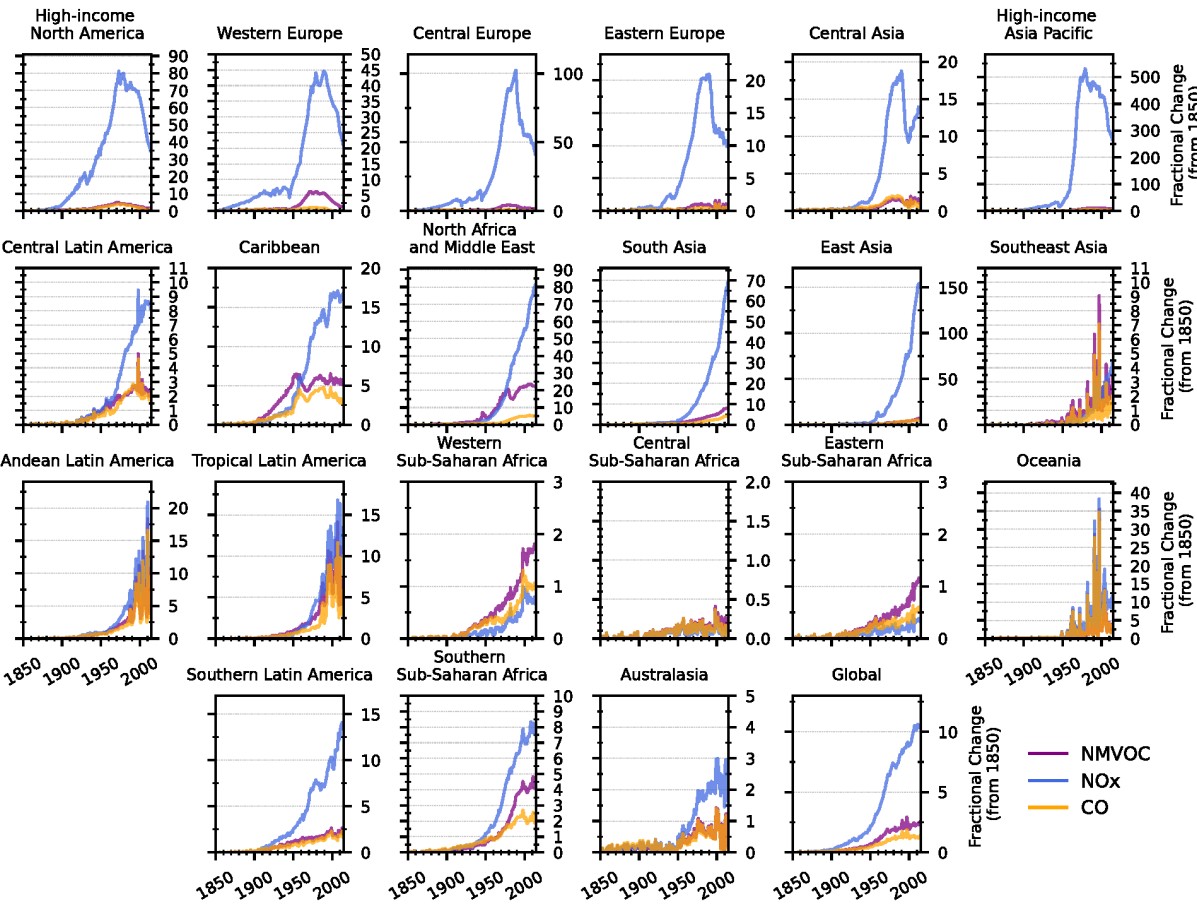

**Figure A3.** Relative change in historical regional emissions of ozone precursors (NOx, CO and non-CH4 VOCs- NMVOCs) compared to 1850 values.

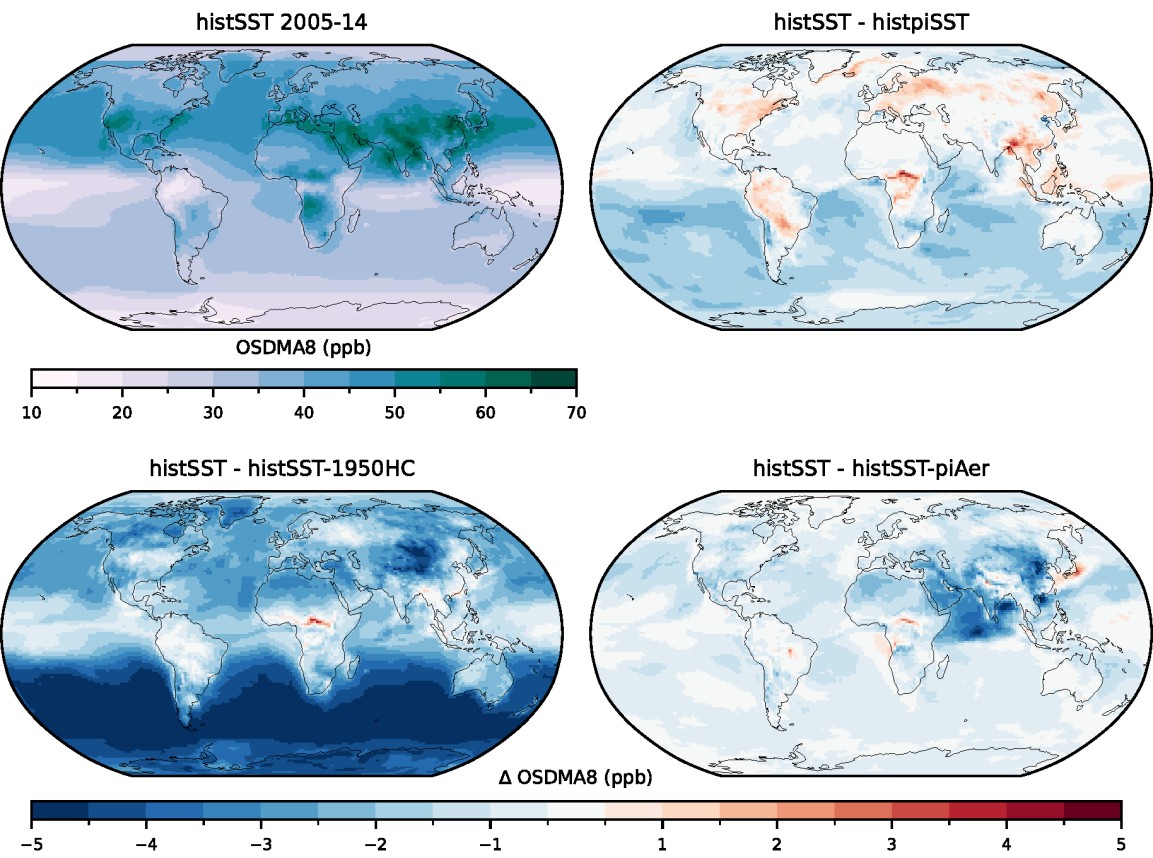

**Figure A4.** 10-year mean in surface OSDMA8 values simulated by UKESM1-0-LL in the HistSST experiment for 2005-14 (bias-corrected) and the change in the same time period in the HistSST experiment relative to the histpiSST, histSST-1950HC and histSST-piAer sensitivity experiments with different drivers of ozone (climate change, stratospheric ozone and aerosols) fixed. Same information as Figure 6 but different colour scale used

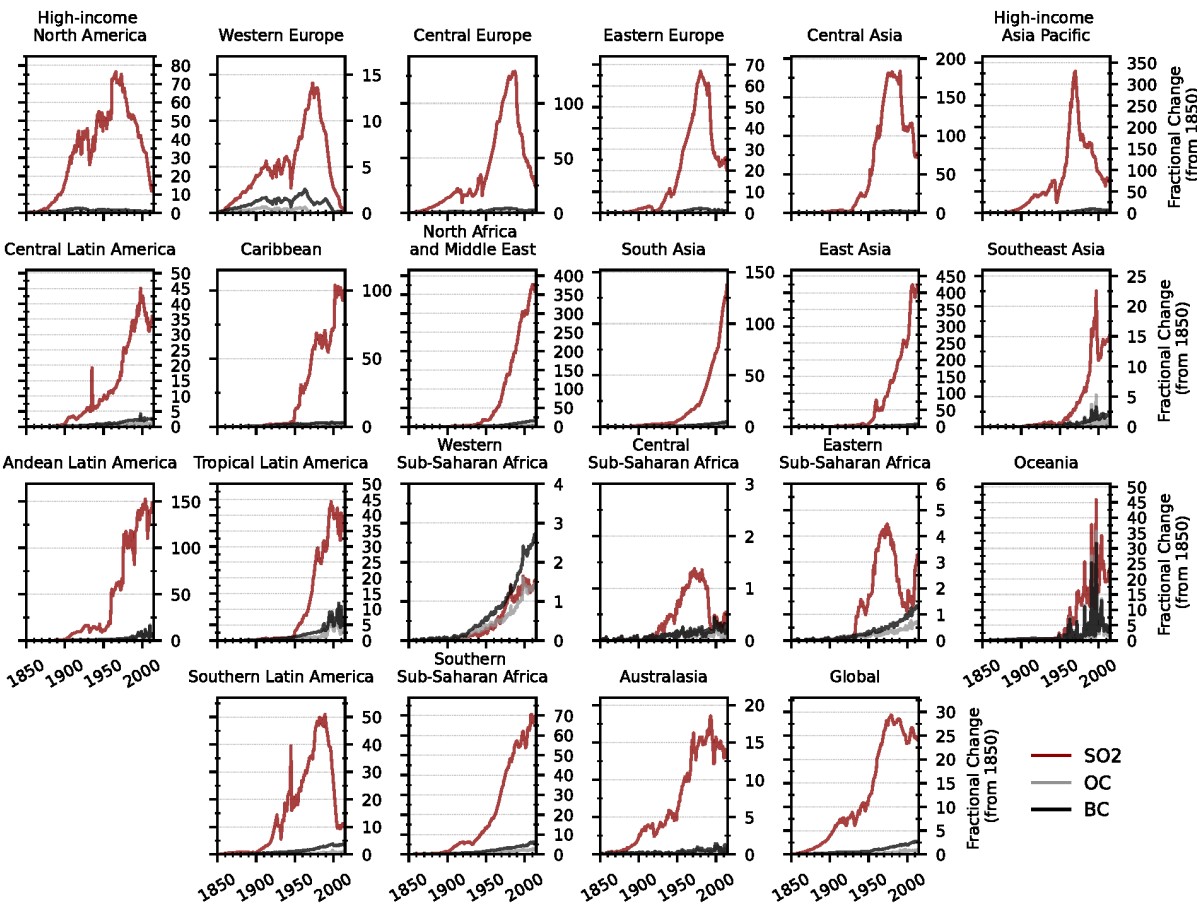

**Figure A5.** Relative change in historical regional emissions of ozone precursors (SO2, OC and BC) compared to 1850 values

**Table A1.** Brief description of Aerosol and Chemistry relevant Processes in the models used in this study

| Model | Resolution | Aerosol Scheme | Chemistry Scheme | Interactive Elements | Reference |
|---|---|---|---|---|---|
| EC-Earth3-AerChem | Approximately $3° \times 2°$ in horizontal (250km) and L34 (0.1hPa) in the vertical | M7 modal aerosol microphysical scheme for $SO_4$, BC, OA, sea salt, and mineral dust in 4 soluble and 3 insoluble modes. $NH_4$, $NO_3$, MSA using Equilibrium Simplified Aerosol Model (EQSAM). | TM5 chemistry scheme accounts for gas-phase, aqueous-phase, and heterogeneous chemistry based on modified version of the CB05 carbon bond mechanism. | Biogenic emissions of VOCs and CO are prescribed using monthly estimates from the MEGAN-MACC data set for the year 2000. Online emissions of mineral dust and sea salt, the oceanic source of DMS, and the production of nitrogen oxides ($NO_X$) by lightning. terrestrial DMS emissions from soils and vegetation, biogenic emissions of $NO_X$ and $NH_3$ from soils, oceanic emissions of CO, NMVOCs and $NH_3$, and $SO_2$ fluxes from continuously emitting volcanoes. | (van Noije et al., 2021) |
| GFDL-ESM4 | Cubed-sphere (c96) grid, with $\approx 100$ km native resolution, regridded to $1.0° \times 1.25°$ and L49 (0.01 hPa) in vertical | Bulk mass-based scheme for $NH_4$, $SO_4$, $NO_3$, BC, OM, sea salt and dust. 5 size bins are used for sea salt and dust. | Interactive stratosphere-troposphere. 43 photolysis reactions, 190 gas-phase kinetic reactions and 15 heterogeneous reactions. $NO_X$-$HO_X$-$O_X$-chemical cycles and CO, $CH_4$ and NMVOC oxidation reactions. | DMS and sea salt emissions calculated online as a function of wind speed (and a prescribed DMS seawater climatology). Dust emissions coupled to interactive vegetation. Lightning $NO_X$ calculated online as a function of convection. Online emissions of BVOCs (isoprene and monoterpenes) calculated from a prescribed vegetation cover using MEGAN2.1 algorithm, which has dependence on light and temperature but also inhibits isoprene emissions based on $CO_2$. | (Horowitz et al., 2020; Dunne et al., 2020) |
| UKESM1-0-LL | $1.25° \times 1.875°$ in horizontal (150km) L85 (85km) in vertical | GLOMAP-Mode. (Modal scheme, mass and number) for $SO_4$, BC, OM, sea salt. Mass based bin scheme used for mineral dust. | UKCA coupled stratosphere-troposphere. Interactive photolysis. 84 chemical tracers. Simulates chemical cycles of $O_X$, $HO_X$ and $NO_X$, as well as oxidation reactions of CO, $CH_4$ and NMVOCs. In addition, heterogeneous processes, Cl and Br chemistry are included. | Online emissions of DMS, sea-salt and dust aerosols, as well as emissions of primary marine organics and biogenic organic compounds. Online $NO_X$ calculated from lightning, Interactive emissions of Isoprene (linked to chemistry) and monoterpenes (linked to secondary aerosols) using light and temperature, but isoprene emissions are inhibited based on $CO_2$. | (Archibald et al., 2020b; Mulcahy et al., 2020) |

*Author contributions.* Steven T. Turnock set out the conceptual idea of the research project with inputs from Dimitris Akritidis and Andrea Pozzer. The main formal analysis, including the creation of figures, was conducted by Steven T. Turnock and Dimitris Akritidis. Observational data and additional analysis on this data was provided by Hantao Wang. Additional model data was provided by Larry Horowitz and Putian Zhou. Steven T. Turnock prepared the manuscript with contributions of writing and editing from all co-authors.

550    *Competing interests.* At least one of the (co-)authors is a member of the editorial board of Atmospheric Chemistry and Physics.

*Acknowledgements.* The contributions of Steven Turnock were funded by the Met Office Climate Science for Service Partnership (CSSP) China project under the International Science Partnerships Fund (ISPF). The contributions from Mariano Mertens were funded by the German Federal Ministry of Education and Research (Funding Nr.: 01LN2207A, IMPAC$^2$T).

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
