# Peer review of "Drivers of change in Peak Season Surface Ozone Concentrations and Impacts on Human Health over the Historical Period (1850-2014)"

_EGUsphere, 2024_

## Community Comment (CC1)

December 18, 2024

Comments by Owen R. Cooper (TOAR Scientific Coordinator of the Community Special Issue) on:

**Drivers of change in Peak Season Surface Ozone Concentrations and Impacts on Human Health over the Historical Period (1850-2014)**

Steven T. Turnock, Dimitris Akritidis, Larry Horowitz, Mariano Mertens, Andrea Pozzer, Carly L. Reddington, Hantao Wang, Putian Zhou, and Fiona O'Connor

EGUsphere [preprint], https://doi.org/10.5194/egusphere-2024-2732
Discussion started Oct. 14, 2024
Discussion closes Dec. 5, 2024

This review is by Owen Cooper, TOAR Scientific Coordinator of the TOAR-II Community Special Issue. I, or a member of the TOAR-II Steering Committee, will post comments on all papers submitted to the TOAR-II Community Special Issue, which is an inter-journal special issue accommodating submissions to six Copernicus journals:  ACP (lead journal), AMT, GMD, ESSD, ASCMO and BG. The primary purpose of these reviews is to identify any discrepancies across the TOAR-II submissions, and to allow the author teams time to address the discrepancies.  Additional comments may be included with the reviews. While O. Cooper and members of the TOAR Steering Committee may post open comments on papers submitted to the TOAR-II Community Special Issue, they are not involved with the decision to accept or reject a paper for publication, which is entirely handled by the journal's editorial team.

**Comments regarding TOAR-II guidelines:**

TOAR-II has produced two guidance documents to help authors develop their manuscripts so that results can be consistently compared across the wide range of studies that will be written for the TOAR-II Community Special Issue.  Both guidance documents can be found on the TOAR-II webpage: https://igacproject.org/activities/TOAR/TOAR-II

*The TOAR-II Community Special Issue Guidelines*:   In the spirit of collaboration and to allow TOAR-II findings to be directly comparable across publications, the TOAR-II Steering Committee has issued this set of guidelines regarding style, units, plotting scales, regional and tropospheric column comparisons, and tropopause definitions.

*The TOAR-II Recommendations for Statistical Analyses*:  The aim of this guidance note is to provide recommendations on best statistical practices and to ensure consistent communication of statistical analysis and associated uncertainty across TOAR publications. The scope includes approaches for reporting trends, a discussion of strengths and weaknesses of commonly used techniques, and calibrated language for the communication of uncertainty. Table 3 of the TOAR-II statistical guidelines provides calibrated language for describing trends and uncertainty, similar to the approach of IPCC, which allows trends to be discussed without having to use the problematic expression, "statistically significant".

**General comments:**

Lines 52-54
This discussion seems to be referring to the suggestion that surface ozone at northern mid-latitudes might have doubled from the 1950s to the year 2000, and that this rapid increase is not reproduced by the models. However, the idea that ozone doubled is now out of date. In the first phase of the Tropospheric Ozone Assessment Report, Tarasick et al. (2019) conducted the most comprehensive assessment of historical ozone observations, digging up old and forgotten datasets missed by other studies.  Tarasick et al. (2019) did not find evidence for a doubling of surface ozone, and concluded that ozone increased by roughly 30-70% from the mid-20$^{th}$ Century until the early 21$^{st}$ Century.  These findings were accepted by IPCC AR6 (Gulev et al., 2021), and the CMIP6 models generally agreed with this rate of increase (Griffiths et al., 2021).

Line 71
A more recent analysis of the ozone climate penalty is provided by Zanis et al., 2022.

Line 194
The cold bias in UKESM is mentioned here, but no explanation is given. Could this be related to higher aerosol loading in this model?

Line 236
I realize that the output is focusing on 10-year periods, but the year 1855 doesn't really coincide with 1850, as listed on line 227.

Line 242
Here it is stated that ozone more than doubled in most regions from 1950 to 2010, but from my reading of Figure 3, the increases were not this great.  As the plots are small, it is not easy to read the numbers exactly, but ozone in South Asia seems to increase from about 30 ppb to nearly 60 ppb, so for this region one could argue for a doubling, but not for the other regions. For example, it seems that ozone in Western Europe increased from 30 ppb to about 44 ppb (an increase of about 50%) from 1950 to 2010, with a similar increase in High-income North America. These 50% increases fall within the 30-70% increase reported by Tarasick et al. (2019), based on historical observation at northern mid-latitudes.

Figure 3 shows that the highest ozone mixing ratios in 1850 occurred in Central Sub-Saharan Africa, and Figure 5 shows that the highest attributable fraction occurred in the same region. What drove ozone production in this region in 1850? Biomass burning?

Figure 4
The concept of the ozone climate penalty indicates that ozone concentrations in high emission regions should increase (if emissions are held constant) due to the increase in heatwave conditions (across highly populated areas) that are more conducive to ozone formation. However, the histpiSST simulation shows that ozone exposure increased slightly when the climate was fixed at 1850. One would think that ozone exposure would decrease if the climate was fixed at the colder 1850 conditions (i.e. less heatwaves in highly populated areas). What is the explanation for the increase?

Line 285
Unites should be United

Line 290
You can't really say that there was no risk globally in 1850, as ozone values in Central Sub-Saharan Africa were above 32.4 ppb.

Line 355
Here it would be helpful to indicate the magnitude of the future temperature changes that produced the notable ozone climate penalty reported by Zanis et al. 2022.

**References**

Griffiths, P. T., Murray, L. T., Zeng, G., Shin, Y. M., Abraham, N. L., Archibald, A. T., Deushi, M., Emmons, L. K., Galbally, I. E., Hassler, B., Horowitz, L. W., Keeble, J., Liu, J., Moeini, O., Naik, V., O'Connor, F. M., Oshima, N., Tarasick, D., Tilmes, S., Turnock, S. T., Wild, O., Young, P. J., and Zanis, P.: Tropospheric ozone in CMIP6 simulations, Atmos. Chem. Phys., 21, 4187–4218, https://doi.org/10.5194/acp-21-4187-2021, 2021.

Gulev, S.K., P.W. Thorne, J. Ahn, F.J. Dentener, C.M. Domingues, S. Gerland, D. Gong, D.S. Kaufman, H.C. Nnamchi, J. Quaas, J.A. Rivera, S. Sathyendranath, S.L. Smith, B. Trewin, K. von Schuckmann, and R.S. Vose, 2021: Changing State of the Climate System. In Climate Change 2021: The Physical Science Basis. Contribution of Working Group I to the Sixth Assessment Report of the Intergovernmental Panel on Climate Change [Masson-Delmotte, V., P. Zhai, A. Pirani, S.L. Connors, C. Péan, S. Berger, N. Caud, Y. Chen, L. Goldfarb, M.I. Gomis, M. Huang, K. Leitzell, E. Lonnoy, J.B.R. Matthews, T.K. Maycock, T. Waterfield, O. Yelekçi, R. Yu, and B. Zhou (eds.)]. Cambridge University Press, Cambridge, United Kingdom and New York, NY, USA, pp. 287–422, doi:10.1017/9781009157896.004

Tarasick, D. W., I. E. Galbally, O. R. Cooper, M. G. Schultz, G. Ancellet, T. Leblanc, T. J. Wallington, J. Ziemke, X. Liu, M. Steinbacher, J. Staehelin, C. Vigouroux, J. W. Hannigan, O. García, G. Foret, P. Zanis, E. Weatherhead, I. Petropavlovskikh, H. Worden, M. Osman, J. Liu, K.-L. Chang, A. Gaudel, M. Lin, M. Granados-Muñoz, A. M. Thompson, S. J. Oltmans, J. Cuesta, G. Dufour, V. Thouret, B. Hassler, T. Trickl and J. L. Neu (2019), Tropospheric Ozone Assessment Report: Tropospheric ozone from 1877 to 2016, observed levels, trends and uncertainties. Elem Sci Anth, 7(1), DOI: http://doi.org/10.1525/elementa.376

Zanis, P., Akritidis, D., Turnock, S., Naik, V., Szopa, S., Georgoulias, A.K., Bauer, S.E., Deushi, M., Horowitz, L.W., Keeble, J. and Le Sager, P., 2022. Climate change penalty and benefit on surface ozone: a global perspective based on CMIP6 earth system models. Environmental Research Letters, 17(2), p.024014.

---

## Author Comment (AC1)

**Author's response to community comments on "Drivers of change in Peak Season Surface Ozone Concentrations and Impacts on Human Health over the Historical Period (1850-2014)"**

Steven T. Turnock et al.

Correspondence to: Steven T. Turnock

(steven.turnock@metoffice.gov.uk)

We would like to thank the TOAR-II community for their helpful and constructive comments on the manuscript. Below we have responded to each comment in turn and made alterations to the manuscript where appropriate (shown enclosed in "*speech marks and italic font*" and any deletions from the manuscript shown with a strikethrough ""). The referee comments are shown first in grey shading and the author's response is shown below in normal font.

This review is by Owen Cooper, TOAR Scientific Coordinator of the TOAR-II Community Special Issue. I, or a member of the TOAR-II Steering Committee, will post comments on all papers submitted to the TOAR-II Community Special Issue, which is an inter-journal special issue accommodating submissions to six Copernicus journals: ACP (lead journal), AMT, GMD, ESSD, ASCMO and BG. The primary purpose of these reviews is to identify any discrepancies across the TOAR-II submissions, and to allow the author teams time to address the discrepancies. Additional comments may be included with the reviews. While O. Cooper and members of the TOAR Steering Committee may post open comments on papers submitted to the TOAR-II Community Special Issue, they are not involved with the decision to accept or reject a paper for publication, which is entirely handled by the journal's editorial team.

**Comments regarding TOAR-II guidelines**:

TOAR-II has produced two guidance documents to help authors develop their manuscripts so that results can be consistently compared across the wide range of studies that will be written for the TOAR-II Community Special Issue. Both guidance documents can be found on the TOAR-II webpage: https://igacproject.org/activities/TOAR/TOAR-II

The TOAR-II Community Special Issue Guidelines: In the spirit of collaboration and to allow TOAR-II findings to be directly comparable across publications, the TOAR-II Steering Committee has issued this set of guidelines regarding style, units, plotting scales, regional and tropospheric column comparisons, and tropopause definitions.

The TOAR-II Recommendations for Statistical Analyses: The aim of this guidance note is to provide recommendations on best statistical practices and to ensure consistent communication of statistical analysis and associated uncertainty across TOAR publications. The scope includes approaches for reporting trends, a discussion of strengths and weaknesses of commonly used techniques, and calibrated language for the communication of uncertainty. Table 3 of the TOAR-II statistical guidelines provides calibrated language for describing trends and uncertainty, similar to the approach of IPCC, which allows trends to be discussed without having to use the problematic expression, "statistically significant".

**General comments:**

Lines 52-54

This discussion seems to be referring to the suggestion that surface ozone at northern mid-latitudes might have doubled from the 1950s to the year 2000, and that this rapid increase is not reproduced by the models. However, the idea that ozone doubled is now out of date. In the first phase of the Tropospheric Ozone Assessment Report, Tarasick et al. (2019) conducted the most comprehensive assessment of historical ozone observations, digging up old and forgotten datasets missed by other studies. Tarasick et al. (2019) did not find evidence for a doubling of surface ozone, and concluded that ozone increased by roughly 30-70% from the mid-20th Century until the early 21st Century. These findings were accepted by IPCC AR6 (Gulev et al., 2021), and the CMIP6 models generally agreed with this rate of increase (Griffiths et al., 2021).

Thanks for the comment on this. However, we specifically refer to the Tarasick et al., (2019) paper and the 30-70% increase in surface ozone over the second half of the 20th Century and into the 21st Century on lines 42 to 43 of the introduction section. We specifically mention that global model simulations tend to agree with this observed range, although might slightly overpredict it across some regions. We however do mention a doubling of surface ozone concentrations and we have reworded this in the abstract and results section to be line with previous studies.

We have also included an additional reference to the Tarasick et al., (2019) results on line 237 as follows:

"*In addition, this simulated historical change in global OSDMA8 is within the observed change of surface ozone across the northern Hemisphere of 30 to 70 % (approximately 10 to 16 ppb) from Tarasick et al., (2019).*"

Line 71

A more recent analysis of the ozone climate penalty is provided by Zanis et al., 2022.

We have Included a reference to the Zanis et al., (2022) paper in the sentence that ends on line 71.

Line 194

The cold bias in UKESM is mentioned here, but no explanation is given. Could this be related to higher aerosol loading in this model?

We have included an additional sentence on line 195 to comment on this as follows:

"*The cold temperature biases simulated by UKESM1-0-LL have been previously attributed to an excessive aerosol forcing, with recent changes to aerosol and cloud properties in an updated version (UKESM1-1-LL) showing an improved representation of historical surface temperatures (Zhang et al., 2021; Mulcahy et al., 2023).*"

Zhang, J., Furtado, K., Turnock, S. T., Mulcahy, J. P., Wilcox, L. J., Booth, B. B., Sexton, D., Wu, T., Zhang, F., and Liu, Q.: The role of anthropogenic aerosols in the anomalous cooling from 1960 to 1990 in the CMIP6 Earth system models, Atmos. Chem. Phys., 21, 18609–18627, https://doi.org/10.5194/acp-21-18609-2021, 2021.

Mulcahy, J. P., Jones, C. G., Rumbold, S. T., Kuhlbrodt, T., Dittus, A. J., Blockley, E. W., Yool, A., Walton, J., Hardacre, C., Andrews, T., Bodas-Salcedo, A., Stringer, M., de Mora, L., Harris, P., Hill, R., Kelley, D., Robertson, E., and Tang, Y.: UKESM1.1: development and evaluation of an updated configuration of the UK

Earth System Model, Geosci. Model Dev., 16, 1569–1600, https://doi.org/10.5194/gmd-16-1569-2023, 2023.

Line 236

I realize that the output is focusing on 10-year periods, but the year 1855 doesn't really coincide with 1850, as listed on line 227.

The population data is only available at the start of each decade in every 10-year period before 2000, which doesn't coincide with the middle of the 10-year mean period that we have used as output from the model. Therefore, we have picked the population at the start of each decadal time-period to be used as representative of the whole 10-year time period.

We have amended the sentences on line 226 to 229 of the manuscript as follows to mention this point:

"*An assessment of population exposure to ozone is also provided by calculating the number of people exposed to OSDMA8 concentrations above the TMREL in 3 different decadal time periods (1850-59, 1980-89 and 2005-14 ). Population data for these periods is obtained for the nearest available years (1850, 1980, 2010) from the Hyde (History database of the Global Environment) dataset (Klein Goldewijk et al., 2017).*"

Line 242

Here it is stated that ozone more than doubled in most regions from 1950 to 2010, but from my reading of Figure 3, the increases were not this great. As the plots are small, it is not easy to read the numbers exactly, but ozone in South Asia seems to increase from about 30 ppb to nearly 60 ppb, so for this region one could argue for a doubling, but not for the other regions. For example, it seems that ozone in Western Europe increased from 30 ppb to about 44 ppb (an increase of about 50%) from 1950 to 2010, with a similar increase in High-income North America. These 50% increases fall within the 30-70% increase reported by Tarasick et al. (2019), based on historical observation at northern mid-latitudes.

Thanks for the comment on this and line 242 to 245 has been amended to:

"*Across most regions OSDMA8 concentrations only show small increases up until about 1950. After this time period concentrations rapidly increase ( by approximately 50%) to 2005-2014 values, driven by the large rapid changes in all anthropogenic precursors of ozone (CH4, NOx, CO, Non-CH4 VOCs), with the largest relative changes occurring in NOx emissions (Fig. 2).*"

Figure 3 shows that the highest ozone mixing ratios in 1850 occurred in Central Sub-Saharan Africa, and Figure 5 shows that the highest attributable fraction occurred in the same region. What drove ozone production in this region in 1850? Biomass burning?

Indeed, ozone production in 1850 is probably likely due to the presence of elevated biomass burning and natural emission sources of ozone precursors (NOx and VOCs) within the model in this time period over this part of Africa.

Figure 4

The concept of the ozone climate penalty indicates that ozone concentrations in high emission regions should increase (if emissions are held constant) due to the increase in heatwave conditions (across highly populated areas) that are more conducive to ozone formation. However, the histpiSST simulation shows that ozone exposure increased slightly when the climate was fixed at 1850. One would think that ozone exposure would decrease if the climate was fixed at the colder 1850 conditions (i.e. less heatwaves in highly populated areas). What is the explanation for the increase?

The comment is correct in that the climate change penalty in figure 4 (comparing the difference of histSST and histpiSST) shows a small reduction in global population exposure to ozone concentrations above the theoretical minimum risk exposure level (32.4ppb). The comparison in Figure 4 does not reflect the actual change in ozone concentrations across regions but rather shows if there is a change in exposure to elevated ozone concentrations which could be harmful to human health. For example, across regions like south and east Asia the regional ozone concentrations remain above 32.4 ppb in both histSST and histpiSST and as such there is no change in population exposure above 32.4 ppb. Figure 4 is meant to highlight if there has been a shift in the population exposure to ozone concentrations in exceedance of 32.4 ppb (and therefore elevated risk to human health) because of the changes in different ozone drivers. Figure 6 and Figure A4 of the manuscript show the actual change in surface ozone due to climate change and highlight the increase in ozone concentrations across continental areas, including populated regions of north America, Europe and Asia.

Line 285

Unites should be United

Changed

Line 290

You can't really say that there was no risk globally in 1850, as ozone values in Central Sub-Saharan Africa were above 32.4 ppb.

Thanks for the comment. A new sentence on line 290 has been included to reflect the elevated concentrations in Central sub-Saharan Africa and non-zero risk to human health in 1850.

"*The major exception to this is that AF values are approximately 5% in the 1850-59 period over the Central Sub-Saharan Africa region. This non-zero AF is due to the elevated pre-industrial OSDMA8 concentrations (Fig 3) arising mainly from biomass burning sources of ozone precursors over this region in this period, although the low population density in this region and time period would represent a low risk to health.*"

Line 355

Here it would be helpful to indicate the magnitude of the future temperature changes that produced the notable ozone climate penalty reported by Zanis et al. 2022.

The range of future temperature change in the study of Zanis (2.7 to 5.4°C by 2100) has now been included in line 355 as follows:

"*The magnitude of the changes in ozone due to 1K of warming over the historical period are consistent with studies analysing the change of ozone in response to future warming (Archibald et al., 2020c; Zanis et al., 2022), with the impact of climate change on surface ozone being less in this study due to the smaller level of historical warming compared to those projected for the future warming of 2.7 to 5.4K.*"

**References**

Griffiths, P. T., Murray, L. T., Zeng, G., Shin, Y. M., Abraham, N. L., Archibald, A. T., Deushi, M., Emmons, L. K., Galbally, I. E., Hassler, B., Horowitz, L. W., Keeble, J., Liu, J., Moeini, O., Naik, V., O'Connor, F. M., Oshima, N., Tarasick, D., Tilmes, S., Turnock, S. T., Wild, O., Young, P. J., and Zanis, P.: Tropospheric ozone in CMIP6 simulations, Atmos. Chem. Phys., 21, 4187–4218, https://doi.org/10.5194/acp-21-4187-2021, 2021.

Gulev, S.K., P.W. Thorne, J. Ahn, F.J. Dentener, C.M. Domingues, S. Gerland, D. Gong, D.S. Kaufman, H.C. Nnamchi, J. Quaas, J.A. Rivera, S. Sathyendranath, S.L. Smith, B. Trewin, K. von Schuckmann, and R.S. Vose, 2021: Changing State of the Climate System. In Climate Change 2021: The Physical Science Basis. Contribution of Working Group I to the Sixth Assessment Report of the Intergovernmental Panel on Climate Change [Masson-Delmotte, V., P. Zhai, A. Pirani, S.L. Connors, C. Péan, S. Berger, N. Caud, Y. Chen, L. Goldfarb, M.I. Gomis, M. Huang, K. Leitzell, E. Lonnoy, J.B.R. Matthews, T.K. Maycock, T. Waterfield, O. Yelekçi, R. Yu, and B. Zhou (eds.)]. Cambridge University Press, Cambridge, United Kingdom and New York, NY, USA, pp. 287–422, doi:10.1017/9781009157896.004

Tarasick, D. W., I. E. Galbally, O. R. Cooper, M. G. Schultz, G. Ancellet, T. Leblanc, T. J. Wallington, J. Ziemke, X. Liu, M. Steinbacher, J. Staehelin, C. Vigouroux, J. W. Hannigan, O. García, G. Foret, P. Zanis, E. Weatherhead, I. Petropavlovskikh, H. Worden, M. Osman, J. Liu, K.-L. Chang, A. Gaudel, M. Lin, M. Granados-Muñoz, A. M. Thompson, S. J. Oltmans, J. Cuesta, G. Dufour, V. Thouret, B. Hassler, T. Trickl and J. L. Neu (2019), Tropospheric Ozone Assessment Report: Tropospheric ozone from 1877 to 2016, observed levels, trends and uncertainties. Elem Sci Anth, 7(1), DOI: http://doi.org/10.1525/elementa.376

Zanis, P., Akritidis, D., Turnock, S., Naik, V., Szopa, S., Georgoulias, A.K., Bauer, S.E., Deushi, M., Horowitz, L.W., Keeble, J. and Le Sager, P., 2022. Climate change penalty and benefit on surface ozone: a global perspective based on CMIP6 earth system models. Environmental Research Letters, 17(2), p.024014.

---

## Author Response (AR1)

**Author's response to reviewer comments on "Drivers of change in Peak Season Surface Ozone Concentrations and Impacts on Human Health over the Historical Period (1850-2014)"**

Steven T. Turnock et al.

Correspondence to: Steven T. Turnock

(steven.turnock@metoffice.gov.uk)

We would like to thank the reviewers for their helpful and constructive comments on the manuscript. Below we have responded to each comment in turn and made alterations to the manuscript where appropriate (shown enclosed in "*speech marks and italic font*" and any deletions from the manuscript shown with a strikethrough ""). The referee comments are shown first in grey shading and the author's response is shown below in normal font.

**Response to Reviewer 1**

General Comment:

This paper argues that increases in NOx emissions and CH4 concentrations have played a major role in the historical changes in surface ozone concentrations. Although many previous studies have already shown that these factors have a significant impact on the historical changes in surface ozone, there have been few studies that have analyzed the latest CMIP6 (AerChemMIP) data from this perspective, so to some extent this result is novel. In addition, the authors analyze the health effects of surface ozone changes. Although these results appear to be novel in the same sense, the interpretation of the results is very simplistic and further analysis and discussion of the results are desirable for publication in the journal. I would like the authors to refer to the following comments for revisions.

Major Comment:

The contribution of each driver to historical changes in OSDMA8 is quantitatively estimated in Section 3.2.1. Is it possible to quantitatively estimate the contribution of each driver to health impact in a similar way? With the current descriptions in the manuscript, it is difficult to consistently compare the contribution of each driver to OSDMA8 and its contribution to health impact. If such a comparison could be made, the difference between the impact of each driver on ozone concentration changes and the health impact would be visualized, the factors behind these differences could be discussed further, and suggestions for future countermeasures would become even more meaningful.

We thank the Reviewer for raising up this point. The ideal approach would be to fully decompose ozone (OSDMA8) changes into the contributions of different drivers and subsequently determine each driver's impact on the ozone health effects. Unfortunately, this is not feasible due to conceptual limitations in the design of the CMIP6 simulations (as concepts like tagging e.g., Grewe et al., (2017); Butler et al., (2018), are not used), the methodology used for the health risk calculations, and the non-linearities present in the contribution of the drivers. First, we should clarify that in this study estimating the effect of different drivers is only possible within a conceptual framework, such as determining what ozone levels in 2010-2014 would have been if NOx emissions had remained at pre-industrial levels. With this approach, we can roughly and primarily qualitatively quantify the contribution of each driver to ozone changes (Section 3.2.1). Nevertheless, we cannot assess the actual percentage contribution of each driver to the total

ozone change over the historical period due to non-linearities in ozone chemical formation processes and other feedbacks. Indicatively, the sum of the individual drivers' contribution is 20% larger than the total ozone change derived from the histSST simulation alone (also referred to the manuscript and directly in the response to specific comments from both reviewer 1 and 2 below). Secondly, the contribution of each driver to the health impact cannot be assessed by simply taking the difference of the attributable fraction (AF) between the histSST and the respective sensitivity calculation, for the following reason. The relative risk calculation (see equation 2 in the manuscript) depends on the exceedance of the OSDMA8 metric above the TMREL threshold of 32.4 ppb. For example, in the near-present period 2010-2014, if OSDMA8 concentrations for the histSST experiment are 60 ppb and for the histSST-piNOx are 32.4 ppb, one could attribute 27.6 ppb (~46%) to NOx emissions. Calculating the AF for the above experiments will result in an RR of 1, thus an AF of 0 for the histSST-piNOx experiment, and a non-zero AF for the histSST experiment. This would erroneously conclude that all health effects in the histSST experiment is due to NOx, which, of course, is not the case. To decompose the AF into individual drivers, one should assume that the percentage contribution of each driver to the OSDMA8 exceedance of TMREL is the same as its contribution to OSDMA8 levels and apply this to the AF. However, this is not feasible, as the actual percentage contribution of each driver cannot be determined for the reason explained earlier. Nevertheless, one should consider the AF for each experiment as a concept, essentially, what the AF would be if NOx emissions were set to pre-industrial levels, to obtain qualitative insights on the AF from individual drivers, like we have done here.

To provide further clarification on the methods we have included the following paragraph in the Subsection 2.3 at line 229 of the original manuscript:

*"Attribution of ozone health effects (AF) to different drivers by taking the difference between the reference and sensitivity simulations is not valid, as AF is not directly proportional to OSDMA8 concentrations but rather depends on whether ozone exceeds the TMREL and by how much. This type of decomposition would be feasible if the actual percentage contribution of each driver to the total OSDMA8 change could be estimated, which is not possible in the current approach due to the non-linearities in ozone chemistry. Therefore, only qualitative insights can be made on the relative contributions of each driver to ozone health effects. Thus, AF in a sensitivity simulation should be viewed as a conceptual indicator, reflecting the potential health impacts under a specific scenario (e.g., pre-industrial NOx emissions in near present-day conditions), without providing a direct attribution of the health effects of ozone to individual drivers in the real-world context."*

According to the previous and considering that the regional average of AF in Figure 7 does not represent exactly the fraction of COPD mortality due to ozone exposure (OSDMA8 is not population-weighted here when used in the calculation of AF) we have modified the paragraph in Subsection 3.2.2 lines 382-393 of original manuscript as follows:

*"Figure 7 presents the global and regional mean AFs  for the histSST and  sensitivity experiments isolating the different drivers of ozone formation in the near present-day period 2015-2014. A high health risk from COPD attributable to ozone pollution (histSST) is found in South Asia and East Asia with a regional average AF of 13% and 12%, respectively; while values in other regions are:  North Africa and Middle East (9%), High-income Asia Pacific (9%), Central Europe (7%), Western Europe (6%) and High-income North America (5%). Regarding the drivers of ozone changes, fixing NOx emissions to pre-industrial levels (histSST-piNOx) eliminates the risk for human health from ozone exposure for the*

*2005-2014 period in most regions.*  *Historical increases in methane concentrations also appear to imply a risk for human health in near present-day both globally and regionally, as setting methane concentrations to pre-industrial levels results in a lower risk for health in the highly populated regions of South Asia, East Asia, and Western Europe with AF values of 9%, 7%, and 2%, respectively.* ~~Historical increases in methane concentrations also appear to be a significant contributor to near present-day AF both globally and regionally. Setting methane concentrations to pre-industrial levels reduces ozone-related mortality in the highly populated regions of South Asia, East Asia and Western Europe by ~4% and that of High-income North America by ~3%.regional mean near present-day AFa small (≈1%) increase in the risk to ozone-related mortality~~."

Specific Comments:

- Table1: It would be better to more clearly explain what is fixed in each sensitivity scenario. Methane is well described in the caption already, but the other drivers are not clearly explained in the table caption.

Thanks for the comment and we have included additional descriptions of the other drivers within caption of Table 1.

- L128: Do all UKESM1 sensitivity experiments use the same initial data?

Yes, the histSST and all the sensitivity experiments use the same initial conditions file.

- L129: heath -> health

Corrected

- L145-147: The bias correction method for the baseline period should be explained in more detail here. Which was used for the correction: the difference or ratio between RAMP and each model? Did the method correct the 1-hour values and then calculate OSDMA8 or correct the OSDMA8 values directly, etc.?

The RAMP dataset directly provides OSDMA8 as the metric for ozone concentrations. Therefore, we have bias corrected the models after calculating this metric and not the "raw" hourly surface ozone concentrations output from the models. We have corrected the model baseline period by the absolute difference between the individual CMIP6 model and the RAMP dataset. We have provided additional details on these points in the methods section as below:

A new line has been included on line 145:

"*The RAMP dataset directly provides ozone concentrations as OSDMA8 values, which are used to correct the corresponding OSDMA8 values simulated by each CMIP6 model.*"

The sentence on line 145 to 147 has been amended to:

"*We then use OSDMA8 values over the last 10 year mean period (2005-2014) from the RAMP dataset to  correct  for the absolute difference in values simulated in same time period  by the CMIP6 models . Using the most recent 10-year mean period of 2005-2014 from the historical experiments  maximises the availability of ground-based observations in the RAMP dataset for correction.*"

The Sentence on line 147 to 149 has been amended to:

"*Figure 1 shows the 10-year mean (2005 to 2014) simulated OSDMA8 values from each CMIP6 model in the histSST experiment and also the absolute difference  compared to the RAMP dataset in the same time period (i.e. the correction applied to each CMIP6 model).*"

Further clarifications on the methodology has also been made in the response to some of the specific comments of reviewer 2 below.

- L175-177: A brief description is desirable here on the ozone response derived with the method of Wild et al. (2012) using the difference between the equilibrium and prescribed CH4 concentration. Whether the ozone increase or decrease? Are there any regional or temporal characteristics?

Thanks to the reviewer for this point and we have tried to provide a more detailed description of the adjustment process in the response to the reviewer below. To avoid confusion and the repetition of unnecessary details in the manuscript we have tried to keep the amount of detail included in the methods section to a minimum.

The ozone response to the difference between the equilibrium and prescribed global $CH_4$ concentrations is calculated by using the relationship derived in Wild et al., (2012) and Turnock et al., (2018) using the results from methane perturbation experiments as part of the HTAP (Hemispheric Transport of Air Pollutants) project. In the HTAP project an experiment was conducted where global methane concentrations were reduced by 20%. The ozone response to such a perturbation in methane was calculated across multiple chemistry models. The results from this multi-model study found that ozone exhibited non-linear responses to changes in methane. From this a relationship was derived to calculate ozone responses to methane perturbations.

The method for adjusting ozone concentrations due to a change in global methane concentrations (such as difference between prescribed and equilibrium concentrations) can be summarised as follows (based on that in Wild et al., 2012 and Turnock et al., 2018).

Firstly, a scale factor for the CH4 response, $f_m$, is calculated as the ratio of the global abundance change, $\Delta CH_4$ (in this study the different between equilibrium and prescribed concentrations), relative to the 20% abundance change in the original HTAP experiments:

$$f_{lin} = \frac{\Delta CH_4}{0.2 \times [CH_4]}$$

A non-linear scaling is then applied to the linear factor calculated above to account for reduced ozone response due to changes from $CH_4$

$$f_{non-lin} = 0.95 f_{lin} + 0.05 f_{lin}^2$$

The ozone response to the change in methane is then calculated by using the non-linear scaling factor ($f_{non-lin}$) to scale the original ozone response from the 20% perturbation HTAP experiment by:

$$\Delta O_3 = f_{non-lin}\Delta O_3$$

This now allows for us to adjust the ozone response in each sensitivity experiment for the change in methane concentrations due to the methane lifetime effect.

The different sensitivity experiments induce a different impact on methane and therefore also ozone. To summarise, four experiments piSST, 1950HC, piNOx and piO3 all result in an increase in methane concentrations and therefore an increase in ozone. Whereas, the remaining three experiments piAer, piCH4 and piVOC all result in a decrease in methane concentration and also ozone concentrations. The sign of adjustment in ozone occurs in the same direction across all regions as the experiments conducted by UKESM1 in this study and those conducted in HTAP to inform the relationship used perturbations in global concentrations of methane. However, there are regional variations in the magnitude of the ozone response, which can be attributed to the differing ozone production sensitivities in the models that form the basis of the relationship derived in Wild et al., (2012).

To include some additional details on the above methods we have amended the manuscript text as follows:

We have inserted the following additional text, which includes a new sentence on line 176, an amended the sentence on line 177 to 179:

"*This relationship was obtained by analysing the ozone response of multiple chemistry transport models in an experiment where the global $CH_4$ concentrations was reduced by 20%; conducted as part of phase 1 and 2 of the Hemispheric Transport of Air Pollutant (HTAP) project. A non-linear relationship is then used to scale this ozone response (calculated from the HTAP 20% global $CH_4$ reduction experiment) by the ratio of the global $CH_4$ abundance change in this study (the different between equilibrium and prescribed concentrations) relative to the 20% abundance change used in the original HTAP experiments. From this we now calculate for each sensitivity experiment the response in surface ozone concentrations that occurs due to the change in $CH_4$ concentrations resulting from the adjustment of $CH_4$ lifetime.*"

The following new sentences included on line 180:

"*Four of the sensitivity experiments (piSST, 1950HC, piNOx and piO3) result in an increase in $CH_4$ concentrations and subsequently a relatively small increase in ozone. Whereas, in the other three sensitivity experiments (piAer, piCH4 and piVOC) there is a small reduction in $CH_4$ and ozone concentrations. The magnitude of adjustment to the ozone response slightly varies across regions in each sensitivity experiment, although the sign of change is the same.*"

- L227: concentration -> remove

Removed

- L240-L242: It would be an overstatement that the CMIP6 models have an ability to simulate long-term change in surface ozone based on a comparison at only five remote locations.

Whilst we agree with the reviewer that representing trends at only five monitoring locations is overstating the ability of models to simulate long-term changes in surface ozone, there is a lack

of available continuous multidecadal observations of surface ozone concentrations with which to compare to model simulations. In Griffiths et al., (2021) the five remote ground-based monitoring locations selected for comparison represent the longest continuous surface ozone observation datasets (>40 years) with which to test the ability of global chemistry climate models to reproduce multi-decadal trends. This provides one of the few comparisons of long-term (multi-decadal) trends in both models and observations. We have slightly changed the emphasis of the sentence on line 204 to 242 to reflect the spatial limitations of this comparison.

"*Previously, CMIP6 models were shown to be able to represent the multi-decadal changes in surface ozone concentrations since 1960 at five remote long-term monitoring  locations around the world (Griffiths et al., 2021). Even though this comparison is spatially limited, due to the availability of monitoring locations with multi-decadal observational records, it does provide a degree of  confidence in the ability of models to simulate long-term changes at specific remote locations, representative of background conditions.*"

- L292: The AF value in Greenland exceeds 10%. Why does it happen where the AF value is quite low at other high latitude regions in the Northern Hemisphere?

The AF values in Greenland exceed 10% because the OSDMA8 values across Greenland are also large in the 2005-2010 period (see first panel on Figure 6). This is due to the nature of the topography of the Greenland ice sheet and its representation in global climate models at coarse resolution. Here the altitude of the lowest model level i.e. representative of the surface, is quite elevated in height, i.e. ~2000m above sea level. Therefore, the ozone concentrations here are not representative of typical surface values for this latitude but can be considered more like free-tropospheric ozone concentrations. As such they will be higher in concentration and produce a larger AF than we would expect. However, due to the low population density of Greenland, especially at this elevation on the ice sheet we would expect a minimal impact to human health here.

An additional sentence has been included on line 287 to explain this as follows:

"*High AF values of >8% also occur over Greenland in 2005-2014, which can be attributed to the presence of the high-altitude ice sheet (~2000m above sea level), meaning that the ozone concentrations are not representative of typical surface values for this latitude but can be considered more like elevated free-tropospheric ozone concentrations.*"

- Figure3: Since the average of the three models is mainly discussed in the manuscript, so the average value should also be included in the figure.

Thank you for the suggestion to improve the Figure. However, as Figure 3 shows ozone changes over different regions in multiple panels then it is difficult to add another line showing the multi-model mean whilst not obscuring another one of the individual model lines. We think that showing the diversity of model response in Figure 3 is of more value than a single multi-model mean line so have decided to keep the lines on the plot the same. Hopefully the reader is able to interpret the multi-model mean easily from the individual model responses. To aid in this we have included values of the multi-model mean for each region for the start and end of the time period on a revised Figure 3.

- L299: The number in brackets (e.g. 37%) needs an explanation.

The explanation for this number is provided earlier in the sentence on line 297 to 298. The 9.6 ppb (37%) increase in global mean OSDMA8 concentrations is the $O_3$ response attributed to the

change in $O_3$ precursors (histSST minus histSST-piO3 = NOx, CO and non-CH4 VOCs) in the 2005-14 mean period.

- L312-L313: In this experiment, the CH4 concentration is set uniformly within the model domain, but the actual CH4 concentration has a relatively clear difference between the Northern and Southern Hemispheres. How do you think setting the uniform CH4 concentration affects the change in ground-level ozone concentration?

We agree with the reviewer that using a global $CH_4$ concentration as a surface boundary condition is a limitation, but this approach is common to the majority of global chemistry climate models used in CMIP6. This does provide a global distribution of $CH_4$, constrained to the boundary condition, allowing it to effect chemistry across each grid cell within the model but lacks the spatial heterogeneity of a model with a fully interactive $CH_4$ cycle (driven by $CH_4$ emissions). In a recent $CH_4$ emission driven configuration of UKESM1 Folberth et al., (2022) showed that a $CH_4$ concentration forced simulation would underestimate surface $CH_4$ concentrations in the Northern Hemisphere and overestimate them in the Southern Hemisphere. Therefore, the results presented for the impact of $CH_4$ could also result in a similar underestimation of surface ozone response in the northern hemisphere overestimation in the southern hemisphere.

We have included the following sentence on this limitation at line 312.

"*This means that the impact of changes in $CH_4$ on surface ozone in this study are spatially limited by the absence of a hemispheric gradient in $CH_4$ concentrations (higher in the northern hemisphere and lower in the southern hemisphere), something that would be provided if using a model with a fully interactive $CH_4$ cycle, including $CH_4$ emissions instead of concentrations (Folberth et al., 2022).*"

- L348: including -> remove

Removed

- L352: Do all models include online calculation of BVOC emissions?

No, not all models include interactive BVOC emission scheme. However, this section is specifically referring to the sensitivity simulations run only by UKESM1, which does have interactive BVOC emissions included. A sentence on line 353 has been amended to:

"*This suggests a small climate impact on natural sources of ozone precursor emissions is simulated by the interactive BVOC emission scheme included within UKESM1-0-LL.*"

In response to reviewer 2 we have also included a new table in the appendix (Table A1) showing the differences between models, which also highlights their treatment of BVOCs.

- L354-L355: The UKESM1-0-LL has a smaller ozone sensitivity per unit temperature change (ppb/K) than other models (Zanis et al. 2022), so I guess it is possible that the impact of climate change on OSDMA8 is underestimated in UKESM1-0-LL model. Further discussion on it is desirable here.

The reviewer is correct that in the study by Zanis et al., (2022), UKESM1 had the smallest ozone sensitivity per unit temperature changes on a global mean basis of -0.79 ppbv °C$^{-1}$. This is compared to -0.85 and -0.94 ppbv °C$^{-1}$ for GFDL-ESM4 and EC-Earth3-AerChem respectively. Therefore, this would suggest that UKESM1 underestimates the climate change impact on global mean surface ozone compared to other models. However, this is a global estimation, which is

dominated by the strength of the remote oceanic ozone response to increasing temperatures. This global number does not show important regional differences in the ozone temperature sensitivity of different models, particularly across populated continental areas which are more important for the consideration of health impacts in this study. In the supplementary material of Zanis et al. (2022). Figure S3 shows the spatial distribution of the regression coefficient representing surface ozone sensitivity to temperatures. It is shown that a robust decline in ozone over remote regions is simulated by all models but that also a spatial diversity of individual models is present for ozone sensitivity to temperature over continental areas due to a number of competing processes (e.g. BVOCs, lightning NOx). Compared to the other 4 models presented in Figure S3, UKESM1 shows on average a more positive sensitivity over North America, Europe, Sub-Saharan Africa and South America, with an approximate average positive sensitivity over Asia. Therefore, there is still considerable differences in the simulated magnitude and sign of the changes in surface ozone due to climate change from global chemistry-climate models.

We have included some additional points of discussions on this topic in the paragraph on lines 343 to 355.

Sentence on line 343 has been modified to:

"*The climate change signal over the historical period has resulted in an approximate 1K increase in global mean surface temperatures simulated by UKESM1-0-LL by the 2005 to 2014 period (Fig. 2), which has resulted in a small reduction in global mean OSDMA8 concentrations of 0.8 ppb (2%) that is consistent with the surface ozone temperature sensitivity of -0.79 ppbv °C$^{-1}$ from Zanis et al., (2022).*"

New sentences have been added after line 355:

"*In addition, Zanis et al., (2022) showed that UKESM1-0-LL had a lower ozone sensitivity per unit temperature change on a global mean basis (-0.79 ppbv °C$^{-1}$) than other CMIP6 models (multi-model mean of -0.93 ppbv °C$^{-1}$), which could mean that climate change impacts on surface ozone are underestimated in this study. However, this global sensitivity number obscures the spatial variation in the sensitivity of surface ozone to temperature (both positive and negative) across different models (Zanis et al., 2022). The model diversity highlights that the surface ozone response to climate change is still uncertain in global chemistry climate models and that both a stronger and weaker response is possible across different regions.*"

- L363: According to the manuscript the sum of individual driver impact on historical OSDMA8 change is 20.4 ppb (8.6+1.5+5.9+0.8+0.8+2.8), and the historical change in OSDMA8 in histSST experiment is 12 ppb (described in L236). I couldn't understand how this number (20% larger) was calculated from these values.

Thanks for raising this issue and hopefully we can provide some clarification. When calculating the sum of the individual drivers, some of the contributions to surface ozone response over the historical period are actually negative and act to reduce the overall change in surface ozone, calculated in the histSST experiment. The change in ozone from these drivers therefore cannot be combined as positive response values as the reviewer has done in the comment, as they act to reduce the surface ozone over the historical period. For example, the impact of climate change (piSST), halocarbons (1950HC) and aerosols (piAer) on surface ozone in the 2005 to 2014 mean period is -0.8 ppb, -2.8 ppb and -0.8ppb respectively. These reductions can then be combined with the increase in surface ozone from changes in methane (piCH4) of +5.9 ppb, NOx (piNOx) of +8.6 ppb and CO/VOCs (piVOC) of +1.5 ppb. This results in a total summed change from the

individual drivers of +11.6 ppb over the historical period. The overall change in global mean surface ozone concentrations simulated by only UKESM1-0-LL is +9.2 ppb over the entire historical period. The 12 ppb change in surface ozone referred to on line 236 is the multi-model mean change in surface ozone simulated by the 3 CMIP6 models. Therefore, the difference between the overall change simulated by UKESM1-0-LL in histSST change and the sum of the changes from the individual drivers is about 20%.

We have added signs to the numbers on the surface ozone reductions to make this clearer in the text and avoid the confusion highlighted by the reviewer.

We have also included the individual historical change in surface ozone concentrations just from the histSST simulation conducted by UKESM1-0-LL for a direct comparison to the changes from individual drivers on line 296.

"*Figure 6 shows the change in OSDMA8 concentrations simulated by UKESM1-0-LL in the present-day time period (2005 to 2014) due to the historical changes in all drivers together (histSST) and due to the individual drivers of ozone formation i.e. histSST minus sensitivity experiment. The overall change in global mean OSDMA8 concentrations simulated by UKESM1-0-LL over the historical period in histSST is 9.2 ppb.*"

Line 361 to 363 has been amended to:

"*However, the linear combination of the historical change in global mean OSDMA8 concentrations, calculated from the individual driver experiments by UKESM1-0-LL (+11.6 ppb), is found to be about 20% larger than that calculated when all drivers are varied simultaneously (+9.2 ppb from histSST).*"

We have also included additional points on this section in response to reviewer 2 below.

- L377: concs?

Replaced with "*concentrations*"

- Figure A1: Southern Sub-Saharan Africa is in the wrong colour on the map.

Thanks for spotting this error. We have now changed colour of the region on Figure A1 and replaced with a new version in the revised manuscript.

**Response to Reviewer 2**

General Comment:

This study provides a valuable analysis of historical surface ozone trends and their health impacts using CMIP6 data, which is particularly significant given the limited number of studies attributing global ozone trends to various drivers using this dataset. Notably, the results align well with findings from previous studies, supporting the potential applicability of CMIP6 models for ozone trend attribution. This consistency reinforces confidence in using CMIP6 data for understanding long-term changes in ozone and their underlying drivers.

Overall, the manuscript topic is highly relevant for ACP, and the results provide novel and insightful contributions. However, the paper could benefit from more in-depth analysis and discussion in many areas. A more detailed exploration of the methodological uncertainties and limitations would strengthen the robustness of the conclusions. Additionally, the discussion could expand on how these findings compare quantitatively to previous studies and how CMIP6 models might improve upon or differ from prior modeling frameworks.

We would like to thank the reviewer for this general comment and also the specific comments below on the manuscript.

In replying to the comments from reviewer 1 and those below from reviewer 2 we have made corrections to the manuscript to improve the description of the methods, and in particular where any uncertainties or limitations exist. We have provided further clarification on the methods used for bias correction and how the model sensitivity experiments have been compared to isolate the contribution of the different drivers to surface ozone. In addition, we have included more information on the differences between models in the form of a new Table in the appendix to show how such processes could impact ozone formation e.g. BVOCs. Furthermore, we have clarified how changes in $CH_4$ are treated in the models and the contribution of this precursor to ozone formation. We have also provided additional comments on the ozone temperature sensitivity of the model and how this might impact the magnitude of the change in surface ozone simulated.

In the revised manuscript, we have also included a comparison of the historical change in surface ozone simulated in this study to observations (where available) and to other global chemistry climate model intercomparison projects, including CMIP5, CMIP6 and the 1st phase of the tropospheric ozone assessment report (TOAR-1). Inclusions have been made in response to the community comment and also in the points to reviewer two below to show that at the large scale our simulated change in surface ozone is consistent with observations and other studies. We have amended the manuscript to also include comparison with other studies that attribute more recent changes in surface ozone primarily to changes in anthropogenic ozone precursors emissions; a similar result to this study. However, quantitative comparison to other such attribution studies are not straight forward due to differences in surface ozone metric and the more recent time period (1980 to 2010) used, compared to the full historical time period (1850 to 2014) used in this study.

This study differs from previous ones as it is the first time that hourly surface ozone data has been made available from global chemistry climate models for simulations cover the entire 165-year historical period (1850 to 2014) with which to calculate ozone exposure metrics (OSDMA8) that are relevant to human health. Coupling this with the inclusion of numerous sensitivity experiments related to ozone formation within the AerChemMIP protocol (Collins et al., 2017), then allows this study to link the historical change in surface ozone, and its drivers, to human

health outcomes. In particular the sensitivity studies conducted with the global chemistry climate model UKESM1 allow for the attribution of many different drivers together including ozone precursors, aerosols, ozone depleting substances and climate change. We have tried to outline how this study adds values to previous global model intercomparison studies in the last paragraph of the introduction and have also now put statements on this in the conclusions section by modifying the lines 399 to 400 as follows.

"*For the first time, we use hourly surface ozone output from three different CMIP6 models, over the 165-year historical period (1850 to 2014), to calculate changes in the peak season ozone metric (OSDMA8), that can relate risks to human health from long-term exposure to surface ozone.*"

Line 420 is amended as follows:

"*An analysis of the drivers of historical changes in OSDMA8 concentrations and risks to human health is provided from hourly surface ozone outputs of sensitivity experiments conducted using a single CMIP6 model that isolate the historical changes in ozone precursors (NOx, CO, non-CH$_4$ VOCs and CH$_4$), anthropogenic aerosols, ozone depleting substances and climate change.*"

Line 449 is amended as follows:

"*The results from these experiments provide a unique opportunity to quantify the long-term changes, and attribution to multiple drivers behind the changes, of an ozone exposure metric relevant to health impacts over a 165-year period from 1850 to 2014.*"

Specific Comments:

- L147: Figure 1 provides a comprehensive visualization of the model-simulated OSDMA8 values and their differences from the RAMP dataset. However, I would suggest including an additional subfigure to explicitly illustrate the spatial distribution of the RAMP dataset. This would provide a clearer and more intuitive comparison between the model results and the observed dataset, helping readers better understand the spatial discrepancies.

  Thanks for the comment. We have included an additional sub figure on Figure 1 of the revised manuscript to show the RAMP observations in addition to the model simulations.

- L155: As mentioned, four different bias correction techniques are available: mean bias, relative bias, delta correction, and quantile mapping. The study chose to apply only the delta correction method. It would strengthen the paper to clarify why delta correction was preferred. Were the other three methods unsuitable for this study's context? Did the correction results from the other methods perform less effectively compared to delta correction? Clarifying this methodological choice would improve the transparency and robustness of the study.

  The selection of the relevant bias correction for this study was based upon the work done in the technical note by Staehle et al., (2024), which assessed the performance of different techniques for the correction of surface ozone in global chemistry-climate models. They evaluated the performance of four different correction techniques for bias correcting model projections of surface ozone that included: mean bias (MB), relative bias (RB), delta correction (DC) and quantile mapping (QM). They found that both the QM and DC methods showed less errors when correcting future projections of ozone in global chemistry climate models than the MB an RB techniques. In addition, the evaluation of

techniques found little difference in the performance of the DC and QM for bias correction of future projection. However, because of the much simpler mathematical formulation of DC compared to QM, Staehle et al., (2024) recommended the use of the DC bias correction method. Given that we are also using projections of surface ozone obtained from global chemistry climate models, we chose to use the DC method based on the recommendations of the evaluation in Staehle et al., (2024). This is also consistent with other recent studies that use the DC method Akritidis et al., (2024) and Pozzer et al., (2024).

We have amended the sentences on lines 155 to 157 to include additional details on the selection of the bias correction methods as follows:

"*Staehle et al., (2024)  evaluated the  performance of four different  techniques for bias correcting surface ozone in global chemistry-climate models. The techniques assessed include: mean bias, relative bias, delta correction and quantile mapping. Staehle et al., (2024) recommended using  the delta correction method for correcting future projections of surface ozone  due to its  lower errors compared to mean and relative bias, and numerical simplicity compared to quantile mapping. Delta correction has also been used as a method in other recent air pollution health studies (Akritidis et al., 2024; Pozzer et al., 2024). Therefore, based on the recommendations of Staehle et al., (2024) and other recent studies, we use the delta correction method to calculate the change in historical ozone concentrations by applying the ratio of change between each decadal mean and the present day (2005-2014) on top of the bias corrected baseline for each model.*"

- L163: After applying bias correction to the histSST experiment, how were the results quantitatively compared with other sensitivity experiments to isolate and measure the specific contributions of individual drivers? A more detailed explanation of the comparison process would be needed.

As explained in the text (lines 159-163), the contribution of each individual driver has been obtained by directly comparing pairs of bias corrected model experiments i.e. a control experiment with a perturbation experiment. Here the control experiment is the historical experiment where all drivers are changing simultaneously (histSST) and the perturbation experiment is an individual sensitivity experiment where an individual driver is fixed (e.g. piO3). The contribution of the individual driver is obtained by taking the difference between surface ozone concentrations in the same 10-year mean time period between each of these paired experiments. For example, to quantify the contribution of NOx emissions to OSDMA8 concentrations in the 2005 to 2014 time period we would subtract the simulated OSDMA8 values from this time period in piNOx (when NOx emissions are fixed at 1850) from histSST in the same time period (when NOx emissions have changed). This would represent the change in OSDMA8 concentrations in the 2005 to 2014 time periods if NOx emissions had increased from 1850 to 2014 (i.e. histSST minus piNOx).

We have expanded the information provided on lines 162 to 163 to clarify how the impact from each driver is calculated, as detailed below:

"*  contribution of  each individual historical driver is obtained through a comparison of different pairs of bias corrected model experiments i.e. a control experiment (histSST) with an individual sensitivity experiment where an individual driver*

*is fixed (e.g. piNOx). The contribution of each individual driver is  calculated by  taking the difference between OSDMA8 concentrations in the same 10-year mean time period between each of these paired experiments. For example, to quantify the contribution of NOx emissions to OSDMA8 concentrations in the 2005 to 2014 time period, the 10-year mean OSDMA8 values in this time period from the piNOx experiment (where NOx emissions are fixed at 1850 values) are subtracted from the values in the same time period of the histSST experiment (where NOx emissions have changed) i.e. histSST minus piNOx. This example quantifies the change in OSDMA8 concentrations in the 2005 to 2014 time period to NOx emissions if they had increased from 1850 to 2014 values. This method is then repeated for each individual driver using the relevant control and sensitivity experiment pairs. .*"

- L164: UKESM1-0-LL processes $CH_4$ inputs differently from the other models. A more detailed comparison of the differences among these models would greatly enhance the analysis.

I am unsure what the reviewer means here as all the models used in this study (including UKESM1-0-LL) use prescribed global $CH_4$ concentrations as input to all the CMIP6 experiments. The paragraph on lines 164 to 182 specifically describe how in the post processing of ozone output from the sensitivity experiments, only conducted by UKESM1-0-LL, we have attempted to correct for the impact of using prescribed global $CH_4$ concentrations as input to the model on ozone from changes in other chemical species i.e. feedback on $CH_4$.

A description of the configuration used by each model related to the use of global methane concentrations is provided in the caption to Table 1. It has also now been included for clarity as an additional sentence on Line 114 as below. We have also included a reference to the new Table A1 in the appendix of the paper highlighting some of the differences between the models (see response to the reviewer comment below).

"*All models used in this study prescribe long-lived greenhouse gases and methane as global annual concentrations. Table A1 shows a brief description of the relevant chemistry and aerosols processes in the three different CMIP6 models.*"

- L187: The stated global increase factors for NOx (>11 times), NMVOCs, CO, and CH4 (>2) since 1850 appear inconsistent with the trends shown in Figure 2. Further clarification is required to reconcile this discrepancy.

The inconsistency arrives because Figure 2 shows fractional change (i.e. Y-X / X) in historical emissions relative to 1850 values but not the ratio of change in historical emissions (i.e. Y / X) as described in the text of the manuscript. We thought that using a ratio of change (or factor) would be easier to communicate the magnitude of emission change in the text. However, we also recognise how using different values in the text and figure might be confusing to the reader. Therefore, we have made the numbers described in the text consistent with those on Figure 2 (i.e. relative change). The following changes have been made to the manuscript to reflect this.

Line 186 to 189 have been amended to:

"*Global anthropogenic emissions of NOx have seen the largest increase of the ozone precursor emissions, a  fractional increase of >10 globally since 1850. Global emissions of non-CH$_4$ VOCs and CO, along with global CH$_4$ concentrations have all increased globally by a  fractional change of >12(i.e. more than doubled) since 1850.*"

Line 308 has been changed to:

"*Historically, global CH$_4$ concentrations, the other major precursor gas to ozone formation, have more than doubled ~~increased by a factor of ~2~~ since 1850 (Fig. 2), which has increased global mean OSDMA8 concentrations by 5.9 ppb (20%) in the 2005 to 2014 period.*"

- L235: Were changes in OSDMA8 concentrations calculated as the absolute difference between 2010 and 1855, or as a trend over 1855–2010 (e.g., 12 ± 2.6 ppb per decade (50% per decade))?

These differences do not represent a trend but are the absolute change between 10-year mean values centred on 1855 and 10 year mean values centred on 2010 (consistent with first and last point shown on Figure 3).

We have amended the sentence on Line 235 to make this clearer

"*Globally, the 10-year multi-model mean (+/- 1 S.D. of 3 model values) OSDMA8 concentrations are simulated to have increased by 12 +/- 2.6 ppb (50% increase)  between 10-year mean values centred on1855 and 2010, which is of similar magnitude to annual mean changes simulated by 6 CMIP6 models (Turnock et al., 2020).*"

The above sentence has been further modified in response to another point below.

- L261: Further details are needed on the differences between the models, particularly in areas where these differences significantly impact ozone formation.

We have now included a new table in the appendix (Table A1) of the revised manuscript and included in this response below for clarity. This shows a summary of the relevant aerosol and chemistry processes within each of the three CMIP6 models. Reference to this table has been included on the revision to line 114 (see response to reviewer point above). The information presented in Table A1 highlights that there are differences between the models in terms of resolution (both horizontal and vertical), chemical mechanisms and interactive components, all of which could have important implication for surface ozone formation.

A new sentence has been included here on line 261 to include a brief mention of difference between models, as well as changes to the existing sentences on line 261 to 262.

"*This diversity between models can be attributed to differences in how they represent chemical processes (see Table A1), particularly relevant for the pre-industrial period are differences in the representation of meteorology/climate, as well as the interactive/natural components such as emissions from biogenic sources (BVOCs), lightning NOx and wetland CH$_4$ (Rowlinson et al., 2020; Wild et al., 2020). In addition, differences in resolution (both horizontal and vertical) and chemical mechanisms between each of the models can also contribute to the highlighted diversity in simulated*

*ozone concentrations (Wild and Prather 2006; Wild et al., 2020). These structural differences between models also impacts  the chemical sensitivity of ozone formation in each model, resulting in the different simulated historical changes in OSDMA8 across the models Turnock et al., (2020)."*

We have also included references to interactive BVOC emissions on line 350 in response to a point by reviewer 1.

**Table A1** – Brief description of Aerosol and Chemistry relevant Processes in the models used in this study

| Model Name | Resolution | Aerosol scheme | Chemistry Scheme | Interactive Elements | References |
|---|---|---|---|---|---|
| EC-Earth3-AerChem | Approximately 3° x 2° in horizontal (250km) And L34 (0.1hPa) in the vertical. | M7 modal aerosol microphysical scheme for $SO_4$, BC, OA, sea salt, and mineral dust in 4 soluble and 3 insoluble modes. NH4, NO3, MSA using Equilibrium Simplified Aerosol Model (EQSAM) | TM5 chemistry scheme accounts for gas-phase, aqueous-phase, and heterogeneous chemistry based on modified version of the CB05 carbon bond mechanism. | Biogenic emissions of VOCs and CO are prescribed using monthly estimates from the MEGAN-MACC data set for the year 2000. Online emissions of mineral dust, sea salt, the oceanic source of DMS, and the production of nitrogen oxides ($NO_x$) by lightning. terrestrial DMS emissions from soils and vegetation, biogenic emissions of NOx and $NH_3$ from soils, oceanic emissions of CO, NMVOCs and $NH_3$, and $SO_2$ fluxes from continuously emitting volcanoes. | van Noije, et al. 2021 |
| GFDL-ESM4 | cubed-sphere (c96) grid, with ~100 km native resolution, regridded to 1.0° x 1.25° and L49 (0.01 hPa) in vertical | Bulk mass-based scheme for $NH_4$, $SO_4$, $NO_3$, BC, OM, sea salt and dust. 5 size bins are used for sea salt and dust. | Interactive stratosphere-troposphere. 43 photolysis reactions, 190 gas-phase kinetic reactions and 15 heterogeneous reactions. NOx-HOx-Ox-chemical cycles and CO, $CH_4$ and NMVOC oxidation reactions | DMS and sea salt emissions calculated online as a function of wind speed (and a prescribed DMS seawater climatology). Dust emissions coupled to interactive vegetation. Lightning NOx calculated online as a function of convection. Online emissions of BVOCs (isoprene and monoterpenes) calculated from a prescribed vegetation cover using MEGAN2.1 algorithm, which has dependence on light and temperature but also inhibits isoprene emissions based on $CO_2$. | (Horowitz et al., 2019; Dunne et al., 2020) |
| UKESM1-0-LL | 1.25° x 1.875° in horizontal (150km) L85 (85km) in vertical | GLOMAP-Mode. (Modal scheme, mass and number) for $SO_4$, BC, OM, sea salt. Mass based bin scheme used for mineral dust. | UKCA coupled stratosphere-troposphere. Interactive photolysis. 84 chemical tracers. Simulates chemical cycles of Ox, HOx and NOx, as well as oxidation reactions of CO, $CH_4$ and NMVOCs. In addition, heterogeneous processes, Cl and Br chemistry are included. | Online emissions of DMS, sea-salt and dust aerosols, as well as emissions of primary marine organics and biogenic organic compounds. Online NOx calculated from lightning, Interactive emissions of Isoprene (linked to chemistry) and monoterpenes (linked to secondary aerosols) using light and temperature, but isoprene emissions are inhibited based on $CO_2$. | (Archibald et al., 2020; Mulcahy et al., 2020) |

- L304: Many studies have reported significant ozone changes in these areas due to anthropogenic emissions (Wang et al., 2022; Zhang et al., 2016;...). It would be valuable to include a comparison with these existing findings.

Thank you for the suggestions, although we notice that the two studies reference here are more concerned with trends in tropospheric ozone than surface ozone, which is the focus of this study. They do also include small references to changes in surface ozone but only

over the last few decades. Whereas our results are specifically comparing changes over the 1850 to 2014 time period. Therefore, direct comparison with these studies is not possible. However, we have included some additional text in the manuscript at this point referencing general qualitative comparisons with these and other studies.

New sentences has been included at the end of the paragraph on line 306:

"*Changes in anthropogenic emissions of ozone precursors, particularly over recent decades have also been shown, as in this study, to be the most important driver behind changes in regional surface ozone concentrations (Parrish et al., 2014; Zhang et al., 2016; Lin et al., 2017; Yan et al., 2018; Wang et al., 2022).*"

*A new sentence has also been included at line 361 as:*

"*This attribution of drivers is in agreement with other studies that showed the reductions in peak season surface ozone over North America and Europe and increases over east and south Asia between 1980 and 2010 were primarily driven by changes in anthropogenic ozone precursor emissions, with much smaller contribution from other factors including changes in $CH_4$ concentrations and climate (Zhang et al., 2016; Yan et al., 2018; Wang et al., 2022 ).*"

We have also included some additional references to the change in surface ozone simulated by other global chemistry climate model intercomparison studies. Lines 234 to 236 have been further amended (on top of what occurred in the early response) as follows:

"*Globally, the 10-year multi-model mean (+/- 1 S.D. of 3 model values) OSDMA8 concentrations are simulated to have increased by 12 +/- 2.6 ppb (50% increase)  between 10-year mean values centred on1855 and 2010, which is of similar magnitude to annual mean changes simulated by 6 CMIP6 models (Turnock et al., 2020) and also other previous global chemistry climate model intercomparison studies (Young et al., 2013).*"

- L310: CH4 is also classified as a VOC, but its longer chemical lifetime allows it to contribute more significantly to ozone formation over larger spatial and temporal scales compared to other VOCs. Could you clarify why it leads to a stronger ozone enhancement effect than other VOCs?

The main reason that the historical change in $CH_4$ leads to a stronger ozone response than that for NMVOCs in these experiments is because $CH_4$ has a larger abundance and is well mixed in the atmosphere. In addition, it has also undergone a larger historical change (1850 to 2014) than NMVOCs. Global $CH_4$ concentrations have increased from 808 ppb in 1850 to 1832 ppb in 2014 in the CMIP6 historical experiments, representing a factor of >2 increase in its global abundance. A fairer comparison of the historical change in $CH_4$ and NMVOCs can be made between changes in global emission totals. Table R1 below compares the change in global total (anthropogenic and biomass burning) emissions of $CH_4$ versus NMVOCs over the 1850 to 2014 time period from the historical emission data used in CMIP6 (van Marle et al., 2017; Hoesly et al., 2018). This shows that global $CH_4$ emissions have increased by a much larger amount over the historical period than NMVOCs. Given this additional emission of $CH_4$ coupled with its longer lifetime in the atmosphere means that it contributes to a larger and more homogeneous increase in ozone formation over the historical period than NMVOCs. This result is also consistent

with the sensitivity of surface ozone response to the perturbation in different ozone precursors including emissions of NOx, CO, NMVOCs and global $CH_4$ concentrations by Wild et al., (2012). Furthermore, it is also consistent with other studies where the contribution of $CH_4$ to changes in tropospheric ozone burden and ozone radiative forcing over the historical period has been found to be larger than from NMVOCs (Stevenson et al., 2013; O'Connor et al., 2021). The impact of NMVOCs on surface ozone could be larger on a more local scale e.g. over East Asia, which could be restricted in our results by the horizontal resolution and limited VOC chemistry included within UKESM1 (Archibald et al., 2020).

**Table R1** – Global total (anthropogenic and biomass burning) emissions between 1850 and 2014 (original dataset from van Marle et al., 2017 and Hoesly et al., 2018)

| Emission | 1850 (Tg yr$^{-1}$) | 2014 (Tg yr$^{-1}$) |
|---|---|---|
| $CH_4$ | 44 | 377 (x8.6 increase) |
| NMVOCs | 67 | 234 (x3.5 increase) |

Base on the above we have amended the sentence on line 310 to 312 to include more information, as follows:

"*The impact of changes in $CH_4$ on surface ozone concentrations tends to be larger than other non-methane VOCs and more globally uniform due to its larger change, in emissions and abundance, over the historical period, the longer chemical lifetime of $CH_4$, its larger abundance in the atmosphere, and that it is input to these model experiments as a global annual mean concentration, rather than gridded emissions.*"

- L321: After the signing of the Montreal Protocol, indicating a leveling off of stratospheric ozone decline, Figure 2 shows a significant increase in total column ozone since 2000. How was it determined that stratospheric ozone continued to decline during 2005–2014?

Thank you to the reviewer for bringing this to our attention. Here, we have not been clear about what is being compared. The comparison should be more clearly identified as being between the 10- year mean period of OSDMA8 concentrations in histSST and the same 10-year mean period in the 1950HC experiments. This comparison shows the impact on surface ozone in the 10-year mean period (2005 to 2014) due the changes in ozone depleting substances. It is not meant to refer to changes over the period 2005 to 2014 but rather the 10-year mean of this period. Further clarification on the comparison methodology has been outlined in the response to the third point of the reviewer 2 specific comments.

We have also made the following changes in this section to make the comparisons over the different time periods clearer.

The sentence on line 324 to 325 has been changed to:

"*Global mean OSDMA8 concentrations reduced by -2.8 ppb (-7%) in the 10-year mean period of 2005 to 2014  due to the smaller amounts of stratospheric ozone from the increased ODS in the histSST experiment compared to the 1950HC experiment in this most recent period.*"

- L345/L332: Please specify whether the percentage change is positive (+) or negative (-).

The percentage change in global mean ozone is negative and the sign of change has been added to this value in the text.

- L357: Why are only the attribution results from UKESM1-0-LL presented here? Are the results from the other two models consistent? Please provide an explanation. Additionally, since these models are all online-coupled, this approach does not eliminate feedback effects between various factors and meteorology. How should the influence of these feedback effects be accounted for in the interpretation of the results?

The reason why the results of the sensitivity study is only focussed on UKESM1-0-LL is that the other two models did not conduct all of the same additional AerChemMIP experiments that UKESM1-0-LL did (see line 116 to 199 for further information on this). Additionally, for some of those these additional experiments that were performed, hourly surface ozone was not provided as an output, and it is not possible to make a comparison across these sensitivity experiments between models, as the OSDMA8 metric cannot be estimated. Therefore, we decided to focus analysis on a multi-model comparison for the histSST experiment and a single model comparison of sensitivity experiments.

On the second point of the reviewers comment about these models being online-coupled, all of the experiments analysed in this study used prescribed sea surface temperatures and sea ice concentrations which make them "atmosphere only" in nature (e.g. histSST). This acts to minimise any temperature feedbacks that might occur from changes in the model experiment. However, as the atmosphere is free to respond to any changes in the different experiments then there is likely to be differences in the meteorology and also impacts on ozone chemistry between experiments (e.g. temperature, clouds, photolysis). We have tried to constrain the impact of any individual meteorological year on the results by comparing results over a 10 year mean period in each experiment. However, it is inevitable that there will be impacts from these changes on the results when comparing the histSST experiment to any of the sensitivity experiments as the atmosphere responds. In addition, because the chemical formation of ozone is non-linear (Wu et al., 2009; Wild et al., 2012), then any changes in the individual experiments will also have non-linear impacts on ozone formation, which will be hard to separate out from any meteorological feedbacks. This was highlighted in the summary paragraph on lines 361 to 364 by the fact that the combination of the change in OSDMA8 from the individual sensitivity experiments is 20% larger than the experiment (histSST) when all drivers are changed in combination. This shows that there are different feedbacks and interactions occurring in each of the individual experiments that come together in a non-linear way in the combined histSST experiment. It also indicates that at a global scale the calculated OSDMA8 responses from each individual driver might be slightly overestimated due to the non-linear combination of the effects from simulated feedbacks and interactions.

We have tried to expand this discussion on this by amending the sentence on line 364-365.

"*This indicates at a global scale that there are  feedbacks and interactions (e.g., clouds or aerosols influencing photolysis rates) influencing ozone formation in each of the individual experiments due to the changes from that particular driver (Gao et al., 2020; Qu et al., 2021). These feedbacks/interactions then have a non-linear effect on  the formation of ozone, reducing the magnitude of change in OSDMA8 at the global scale*

*when the drivers are changed in combination in single experiment (histSST) . At the global scale, the impact on ozone concentrations due to each individual driver can be considered overestimated as change in the combined driver experiment (histSST) is smaller than those from the linear addition of the individual driver experiments."*

**References**

Akritidis, D., Bacer, S., Zanis, P., Georgoulias, A. K., Chowdhury, S., Horowitz, L. W., Naik, V., O'Connor, F. M., Keeble, J., Sager, P. L., van Noije, T., Zhou, P., Turnock, S., West, J. J., Lelieveld, J., and Pozzer, A.: Strong increase in mortality attributable to ozone pollution under a climate change and demographic scenario, Environmental Research Letters, 19, 024 041, https://doi.org/10.1088/1748-9326/ad2162,2024.

Archibald, A. T., O'Connor, F. M., Abraham, N. L., Archer-Nicholls, S., Chipperfield, M. P., Dalvi, M., Folberth, G. A., Dennison, F., Dhomse, S. S., Griffiths, P. T., Hardacre, C., Hewitt, A. J., Hill, R. S., Johnson, C. E., Keeble, J., Köhler, M. O., Morgenstern, O., Mulcahy, J. P., Ordóñez, C., Pope, R. J., Rumbold, S. T., Russo, M. R., Savage, N. H., Sellar, A., Stringer, M., Turnock, S. T., Wild, O., and Zeng, G.: Description and evaluation of the UKCA stratosphere–troposphere chemistry scheme (StratTrop vn 1.0) implemented in UKESM1, Geosci. Model Dev., 13, 1223–1266, https://doi.org/10.5194/gmd-13-1223-2020, 2020.

Butler, T., Lupascu, A., Coates, J., and Zhu, S.: TOAST 1.0: Tropospheric Ozone Attribution of Sources with Tagging for CESM 1.2.2, Geosci. Model Dev., 11, 2825–2840, https://doi.org/10.5194/gmd-11-2825-2018, 2018.

Collins, W. J., Lamarque, J.-F., Schulz, M., Boucher, O., Eyring, V., Hegglin, M. I., Maycock, A., Myhre, G., Prather, M., Shindell, D., and Smith, S. J.: AerChemMIP: quantifying the effects of chemistry and aerosols in CMIP6, Geosci. Model Dev., 10, 585–607, https://doi.org/10.5194/gmd-10-585-2017, 2017.

Folberth, G. A., Staniaszek, Z., Archibald, A. T., Gedney, N., Griffiths, P. T., Jones, C. D., et al. (2022). Description and evaluation of an emission-driven and fully coupled methane cycle in UKESM1. Journal of Advances in Modeling Earth Systems, 14, e2021MS002982. https://doi.org/10.1029/2021MS002982

Grewe, V., Tsati, E., Mertens, M., Frömming, C., and Jöckel, P.: Contribution of emissions to concentrations: the TAGGING 1.0 submodel based on the Modular Earth Submodel System (MESSy 2.52), Geosci. Model Dev., 10, 2615–2633, https://doi.org/10.5194/gmd-10-2615-2017, 2017.

Hoesly, R. M., Smith, S. J., Feng, L., Klimont, Z., Janssens-Maenhout, G., Pitkanen, T., Seibert, J. J., Vu, L., Andres, R. J., Bolt, R. M., Bond, T. C., Dawidowski, L., Kholod, N., Kurokawa, J.-I., Li, M., Liu, L., Lu, Z., Moura, M. C. P., O'Rourke, P. R., and Zhang, Q.: Historical (1750–2014) anthropogenic emissions of reactive gases and aerosols from the Community Emissions Data System (CEDS), Geosci. Model Dev., 11, 369–408, https://doi.org/10.5194/gmd-11-369-2018, 2018.

Lin, M., Horowitz, L. W., Payton, R., Fiore, A. M., and Tonnesen, G.: US surface ozone trends and extremes from 1980 to 2014: quantifying the roles of rising Asian emissions, domestic controls,

wildfires, and climate, Atmos. Chem. Phys., 17, 2943–2970, https://doi.org/10.5194/acp-17-2943-2017, 2017.

O'Connor, F. M., Abraham, N. L., Dalvi, M., Folberth, G. A., Griffiths, P. T., Hardacre, C., Johnson, B. T., Kahana, R., Keeble, J., Kim, B., Morgenstern, O., Mulcahy, J. P., Richardson, M., Robertson, E., Seo, J., Shim, S., Teixeira, J. C., Turnock, S. T., Williams, J., Wiltshire, A. J., Woodward, S., and Zeng, G.: Assessment of pre-industrial to present-day anthropogenic climate forcing in UKESM1, Atmos. Chem. Phys., 21, 1211–1243, https://doi.org/10.5194/acp-21-1211-2021, 2021.

Pozzer, A., Steffens, B., Proestos, Y. *et al*. Atmospheric health burden across the century and the accelerating impact of temperature compared to pollution. *Nat Commun* **15**, 9379 (2024). https://doi.org/10.1038/s41467-024-53649-9

Rowlinson, M. J., Rap, A., Hamilton, D. S., Pope, R. J., Hantson, S., Arnold, S. R., Kaplan, J. O., Arneth, A., Chipperfield, M. P., Forster, P. M., and Nieradzik, L.: Tropospheric ozone radiative forcing uncertainty due to pre-industrial fire and biogenic emissions, Atmos. Chem. Phys., 20, 10937–10951, https://doi.org/10.5194/acp-20-10937-2020, 2020.

Staehle, C., Rieder, H. E., Fiore, A. M., and Schnell, J. L.: Technical note: An assessment of the performance of statistical bias correction techniques for global chemistry–climate model surface ozone fields, Atmospheric Chemistry and Physics, 24, 5953–5969, https://doi.org/10.5194/acp-24-5953-2024, 2024.

Stevenson, D. S., Young, P. J., Naik, V., Lamarque, J.-F., Shindell, D. T., Voulgarakis, A., Skeie, R. B., Dalsoren, S. B., Myhre, G., Berntsen, T. K., Folberth, G. A., Rumbold, S. T., Collins, W. J., MacKenzie, I. A., Doherty, R. M., Zeng, G., van Noije, T. P. C., Strunk, A., Bergmann, D., Cameron-Smith, P., Plummer, D. A., Strode, S. A., Horowitz , L., Lee, Y. H., Szopa, S., Sudo, K., Nagashima , T., Josse, B., Cionni, I., Righi, M., Eyring, V., Conley, A., Bowman, K. W., Wild, O., and Archibald, A.: Tropospheric ozone changes, radiative forcing and attribution to emissions in the Atmospheric Chemistry and Climate Model Intercomparison Project (ACCMIP), Atmos. Chem. Phys., 13, 3063–3085, https://doi.org/10.5194/acp-13-3063-2013, 2013.

Turnock, S. T., Wild, O., Dentener, F. J., Davila, Y., Emmons, L. K., Flemming, J., Folberth, G. A., Henze, D. K., Jonson, J. E., Keating, T. J., Kengo, S., Lin, M., Lund, M., Tilmes, S., and O'Connor, F. M.: The impact of future emission policies on tropospheric ozone using a parameterised approach, Atmos. Chem. Phys., 18, 8953–8978, https://doi.org/10.5194/acp-18-8953-2018, 2018.

van Marle, M. J. E., Kloster, S., Magi, B. I., Marlon, J. R., Daniau, A.-L., Field, R. D., Arneth, A., Forrest, M., Hantson, S., Kehrwald, N. M., Knorr, W., Lasslop, G., Li, F., Mangeon, S., Yue, C., Kaiser, J. W., and van der Werf, G. R.: Historic global biomass burning emissions for CMIP6 (BB4CMIP) based on merging satellite observations with proxies and fire models (1750–2015), Geosci. Model Dev., 10, 3329–3357, https://doi.org/10.5194/gmd-10-3329-2017, 2017.

Wang, H., Lu, X., Jacob, D. J., Cooper, O. R., Chang, K.-L., Li, K., Gao, M., Liu, Y., Sheng, B., Wu, K., Wu, T., Zhang, J., Sauvage, B., Nédélec, P., Blot, R., and Fan, S.: Global tropospheric ozone trends, attributions, and radiative impacts in 1995–2017: an integrated analysis using aircraft (IAGOS) observations, ozonesonde, and multi-decadal chemical model simulations, Atmos. Chem. Phys., 22, 13753–13782, https://doi.org/10.5194/acp-22-13753-2022, 2022.

Wild, O., and M. J. Prather (2006), Global tropospheric ozone modeling: Quantifying errors due to grid resolution, J. Geophys. Res., 111, D11305, doi:10.1029/2005JD006605.

Wild, O., Fiore, A. M., Shindell, D. T., Doherty, R. M., Collins, W. J., Dentener, F. J., Schultz, M. G., Gong, S., MacKenzie, I. A., Zeng, G., Hess, P., Duncan, B. N., Bergmann, D. J., Szopa, S., Jonson, J. E., Keating, T. J., and Zuber, A.: Modelling future changes in surface ozone: a parameterized approach, Atmos. Chem. Phys., 12, 2037–2054, https://doi.org/10.5194/acp-12-2037-2012, 2012.

Wild, O., Voulgarakis, A., O'Connor, F., Lamarque, J.-F., Ryan, E. M., and Lee, L.: Global sensitivity analysis of chemistry–climate model budgets of tropospheric ozone and OH: exploring model diversity, Atmos. Chem. Phys., 20, 4047–4058, https://doi.org/10.5194/acp-20-4047-2020, 2020.

Wu, S., B. N. Duncan, D. J. Jacob, A. M. Fiore, and O. Wild (2009), Chemical nonlinearities in relating intercontinental ozone pollution to anthropogenic emissions, *Geophys. Res. Lett.*, 36, L05806, doi:10.1029/2008GL036607.

Yan, Y., Pozzer, A., Ojha, N., Lin, J., and Lelieveld, J.: Analysis of European ozone trends in the period 1995–2014, Atmos. Chem. Phys., 18, 5589–5605, https://doi.org/10.5194/acp-18-5589-2018, 2018.

Zanis, P., Akritidis, D., Turnock, S., Naik, V., Szopa, S., Georgoulias, A. K., Bauer, S. E., Deushi, M., Horowitz, L.W., Keeble, J., Sager, P. L., O'Connor, F. M., Oshima, N., Tsigaridis, K., and van Noije, T.: Climate change penalty and benefit on surface ozone: A global perspective based on CMIP6 earth system models, Environmental Research Letters, 17, 024 014, https://doi.org/10.1088/1748-9326/ac4a34, 2022.

Zhang, Y., Cooper, O. R., Gaudel, A., Thompson, A. M., Nédélec, P., Ogino, S.-Y., and West, J. J.: Tropospheric ozone change from 1980 to 2010 dominated by equatorward redistribution of emissions, Nat. Geosci., 9, 875–879, https://doi.org/10.1038/ngeo2827, 2016.